# Default mode-visual network hypoconnectivity in an autism subtype with pronounced social visual engagement difficulties

Michael V Lombardo[1,2]*, Lisa Eyler[3,4], Adrienne Moore[5], Michael Datko[5], Cynthia Carter Barnes[5], Debra Cha[5], Eric Courchesne[5], Karen Pierce[5]*

[1]Laboratory for Autism and Neurodevelopmental Disorders, Center for Neuroscience and Cognitive Systems @UniTn, Istituto Italiano di Tecnologia, Rovereto, Italy; [2]Autism Research Centre, Department of Psychiatry, University of Cambridge, Cambridge, United Kingdom; [3]Department of Psychiatry, University of California, San Diego, San Diego, United States; [4]VISN 22 Mental Illness Research, Education, and Clinical Center, VA San Diego Healthcare System, San Diego, United States; [5]Autism Center of Excellence, Department of Neuroscience, University of California, San Diego, San Diego, United States

**Abstract** Social visual engagement difficulties are hallmark early signs of autism (ASD) and are easily quantified using eye tracking methods. However, it is unclear how these difficulties are linked to atypical early functional brain organization in ASD. With resting state fMRI data in a large sample of ASD toddlers and other non-ASD comparison groups, we find ASD-related functional hypoconnectivity between 'social brain' circuitry such as the default mode network (DMN) and visual and attention networks. An eye tracking-identified ASD subtype with pronounced early social visual engagement difficulties (GeoPref ASD) is characterized by marked DMN-occipito-temporal cortex (OTC) hypoconnectivity. Increased DMN-OTC hypoconnectivity is also related to increased severity of social-communication difficulties, but only in GeoPref ASD. Early and pronounced social-visual circuit hypoconnectivity is a key underlying neurobiological feature describing GeoPref ASD and may be critical for future social-communicative development and represent new treatment targets for early intervention in these individuals.

**\*For correspondence:**
mvlombardo@gmail.com (MVL);
kpierce@ucsd.edu (KP)

## Introduction

Social visual engagement difficulties, defined as lack of preference for social stimuli often combined with a strong preference for and attention towards non-social stimuli (*Dawson et al., 1998*; *Dawson et al., 2004*; *Klin et al., 2015*; *Chawarska et al., 2013*; *Falck-Ytter et al., 2013*; *Klin et al., 2009*; *Nakano et al., 2010*; *Shic et al., 2011*; *von Hofsten et al., 2009*; *Pierce et al., 2011a*; *Pierce et al., 2016a*; *Jones and Klin, 2013*; *Falck-Ytter et al., 2018*), are key early developmental features of autism spectrum disorders (ASD). These difficulties are central in early ASD screening and diagnostic tools (*Lord et al., 2000*; *Pierce et al., 2011b*; *Robins et al., 2001*). A child's preferences and attention early in life can potentially have large impact on future development and outcome (*Klin et al., 2015*). Reduced social visual engagement behaviors actively select and/or neglect specific types of information from the environment as input to the developing brain. A continual stream of atypical non-social input due to reduced social visual engagement in ASD may have detrimental impact on experience-expectant and experience-dependent processes (*Greenough et al., 1987*; *Holtmaat and Svoboda, 2009*; *Huttenlocher, 2002*) that help to sculpt

**eLife digest** Many parents of children with autism spectrum disorder (ASD) spot the first signs when their child is still a toddler, by noticing that their child is less interested than other toddlers in people and in social play. These early differences in behavior can have long-term implications for brain development. The brains of toddlers with little interest in social stimuli will receive less social input than those of other toddlers. This will make it even harder for the brain to develop the circuits required to support social skills.

But even among children with ASD, there are large differences in children's interest in the social world. One way of measuring these differences is to track eye movements. Lombardo et al. presented toddlers with and without ASD with images of moving colorful geometric shapes next to videos of dancing children. The majority of toddlers, including most of those with ASD, spent more time looking at the children than the shapes. But about 20% of the toddlers with ASD spent most of their time looking at the shapes. These toddlers also had the most severe social symptoms.

To find out why, Lombardo et al. measured the toddlers' brain activity while they slept. During sleep, or when at rest, the brain shows stereotyped patterns of activity. Groups of brain regions that work together – such as those involved in vision – fire in synchrony. Lombardo et al. found that toddlers who preferred looking at shapes over people showed different patterns of brain activity while asleep compared to other children. In the toddlers who preferred shapes, brain networks involved in social skills were less likely to coordinate their activity with networks that support vision and attention.

These findings suggest there may be multiple subtypes of ASD, with different symptoms resulting from different patterns of brain activity. At present, all children who receive a diagnosis of ASD receive much the same behavioral therapy. But in the future, studies of brain networks could allow children to receive more specific diagnoses. This could in turn lead to more effective and personalized treatments.

functional specialization and development in the social brain (*Klin et al., 2015*; *Klin et al., 2003*; *Mundy et al., 2009*; *Johnson et al., 2015*; *Johnson, 2017*). These features are also highly relevant for early intervention. Many early interventions that show success for some individuals (*Bacon et al., 2014*; *Dawson et al., 2010*; *Kasari et al., 2006*; *Pickles et al., 2016*) hinge critically upon the idea of changing this attribute of early ASD development. The hope is that early intervention will increase engagement between the child and the social world and enable experience-dependent neuroplasticity to divert a child towards more typical developmental trajectories (*Dawson, 2008*).

Eye tracking studies of children and adults with ASD have been central in quantifying deficits in social visual engagement (*Klin et al., 2009*; *Nakano et al., 2010*; *Shic et al., 2011*; *von Hofsten et al., 2009*; *Pierce et al., 2011a*; *Pierce et al., 2016a*; *Jones and Klin, 2013*; *Falck-Ytter et al., 2018*). Nonetheless, considerable heterogeneity exists across ASD individuals and the early-age neural bases explaining such features and their developmental variability are not well understood (*Chita-Tegmark, 2016*; *Guillon et al., 2014*). Disentangling early-age heterogeneity is the foundation for making significant progress towards the goals of stratified psychiatry and precision medicine (*Collins and Varmus, 2015*; *Kapur et al., 2012*; *Lai et al., 2013*; *Lombardo et al., 2019*) for ASD and is therefore one of the biggest challenges in the field. Attempts to identify strong neural underpinnings behind ASD are hindered by mixing potentially different ASD subtypes with different biology (*Lombardo et al., 2015*; *Lombardo et al., 2018a*). Furthermore, the clinical relevance of parsing heterogeneity into discrete subtypes is also highly salient - not all individuals take the same developmental path or have similar outcomes (*Lord et al., 2015*) and individuals may vary considerably in responsiveness to early intervention (*Bacon et al., 2014*).

We recently discovered (*Pierce et al., 2011a*) and then replicated (*Pierce et al., 2016a*) the finding of one such clinically relevant subtype with marked lack of early social visual engagement. This subtype can be identified and the deficit objectively quantified with a novel eye tracking preferential looking paradigm, the GeoPref Test (*Figure 1A*). The paradigm displays dynamic non-social colorful geometric patterns side-by-side with social images of happy children in motion. A subset of toddlers comprising around 20% of the ASD population, spend less than 30% of task time looking at social

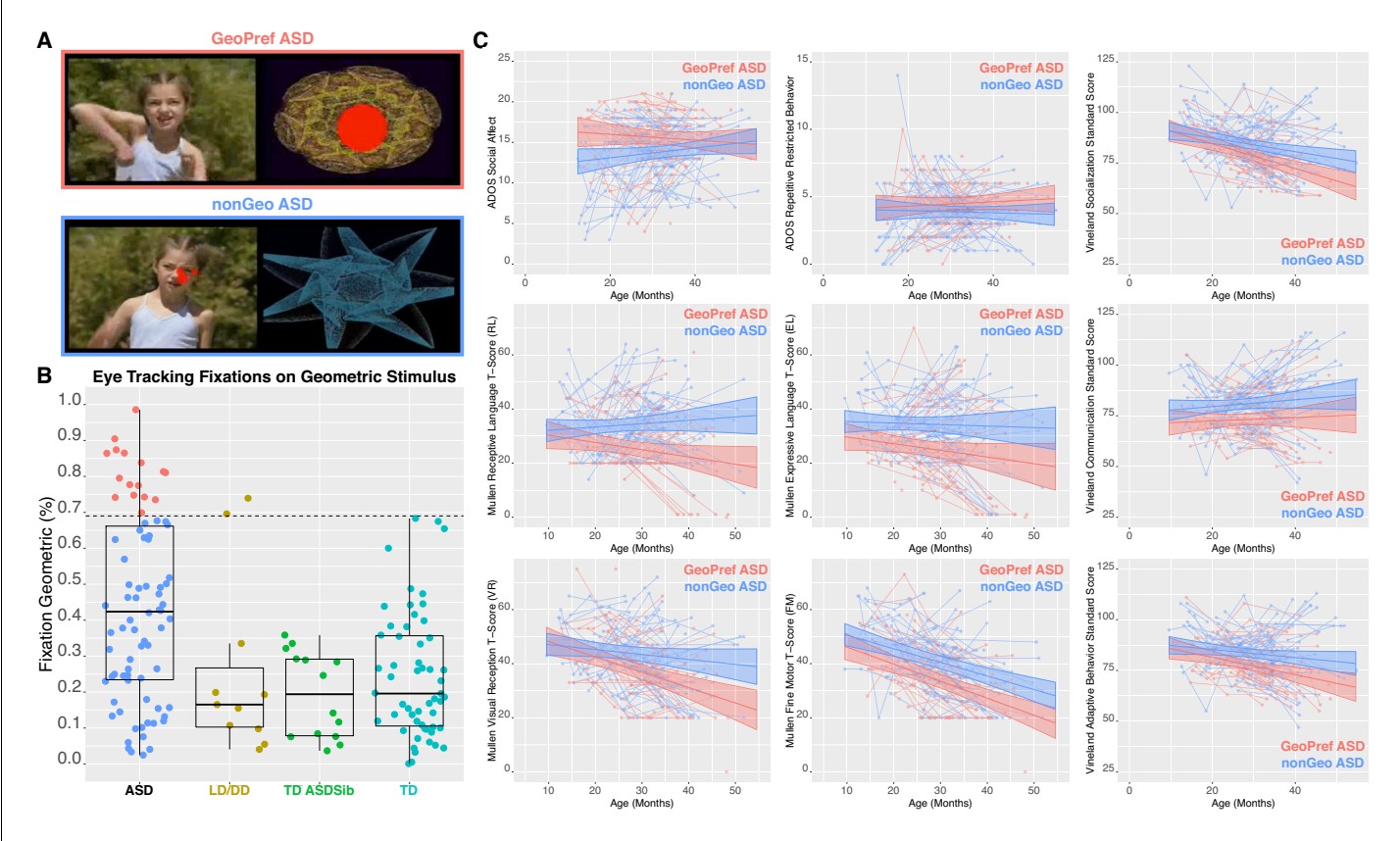

**Figure 1.** Identification of the GeoPref ASD subtype and behavioral differentiation in ASD symptoms, verbal, non-verbal, and adaptive behavior domains. Panel **A** shows examples of the stimuli used in the GeoPref eye tracking test as well as example fixations from a GeoPref ASD individual (pink), and a nonGeo ASD individual (blue). The red dots superimposed on the stimulus show visual fixations and the size of the red dots indicate fixation duration. Panel **B** shows a scatter-boxplot of eye tracking data on the GeoPref test for subjects who also had rsfMRI data available (GeoPref ASD, n = 16, pink; nonGeo ASD, n = 62, blue; language/developmental delay, LD/DD, n = 15 yellow; typically developing siblings of ASD individuals, TD ASDSib, n = 16 green; typically developing toddlers, n = 55, turquoise). The middle line of the boxplot represents the median. The box boundaries represent the interquartile range (IQR; Q1 = 25th percentile, Q3 = 75th percentile), while the whiskers indicate the a distance of 1.5*IQR. Percentage of time fixating on the geometric visual stimulus is plotted on the y-axis and group membership is plotted on the x-axis. The cutoff threshold of 69% is noted as the dashed line. GeoPref ASD toddlers (pink) fall above the cutoff, while all other ASD toddlers (nonGeo ASD; blue) fall below the cutoff. Panel **C** depicts individual and group-level developmental trajectories for longitudinal data from GeoPref ASD (n = 60, pink) or nonGeo ASD (n = 62, blue) on ADOS, Mullen Early Scales of Learning, and Vineland Adaptive Behavior subscales. All measures show a significant main effect of subtype passing FDR q < 0.05. Mullen Receptive Language and Visual Reception subscales additionally show significant (FDR q < 0.05) age*subtype interactions, indicative of different developmental trajectories between the subtypes. The image of a child shown in panel A is taken from a commercially available video (Yoga Kids 3; Gaiam, Boulder, Colorado, http://www.gaiam.com, created by Marsha Wenig, http://yogakids.com) and reproduced here with permission.

© 2003 Gaiam Americas, Inc. All Rights Reserved. *Figure 1A* is taken from a commercially available video (Yoga Kids 3; Gaiam, Boulder, Colorado, http://www.gaiam.com, created by Marsha Wenig, http://yogakids.com) and re-produced here with permission.

The online version of this article includes the following figure supplement(s) for figure 1:

**Figure supplement 1.** Developmental trajectories across Vineland Daily Living Skills and Motor subscales.

displays and instead prefer to attend to geometric patterns 70–100% of the task time. This degree of marked lack of social visual engagement is seldom seen in non-ASD comparison groups, displaying 98% specificity (*Pierce et al., 2016a*). Thus, this unique subtype of ASD toddlers, referred to as 'GeoPref ASD', displays a specific and extreme lack of preference for socially compelling stimuli. GeoPref ASD was originally identified in early screening population-based samples that have high generalizability across the ASD spectrum (*Pierce et al., 2011a*). In the largest eye tracking study of ASD to date (*Pierce et al., 2016a*), we replicated in a large independent sample (n = 334) the same ASD subtype using the original cutoffs derived in our first discovery on this topic (*Pierce et al.,*

*2011a*). GeoPref ASD toddlers are also more severe on a variety of other clinical behavioral measures, indicating that this subtype is highly clinically relevant beyond social visual engagement (*Pierce et al., 2016a*). The GeoPref Test has high test-retest reliability (*Pierce et al., 2016a*) and is simple, fast, and easy to implement, making it a robust behavioral assay for identification of a clinically highly relevant early ASD subtype.

In the current work, we aimed to identify how intrinsic functional connectivity between neural circuits, as measured by resting state fMRI (rsfMRI), is affected in early ASD development and whether heterogeneity in early social visual engagement is a key factor explaining such connectivity differences. Several networks were examined that have high relevance for early social visual engagement. Primary visual cortex, visual association cortices, and networks involved in attentional or salience processing are of high relevance, given their importance in hierarchical processing of features from social visual stimuli (*Haxby et al., 2001*; *Kriegeskorte et al., 2008*; *Uddin et al., 2013*; *Mottron et al., 2006*; *Felleman and Van Essen, 1991*; *Yang et al., 2015*). Subcortical areas such as the amygdala and ventral striatum are also of relevance given theories about social motivation as a key driver of social engagement difficulties in autism (*Chevallier et al., 2012*; *Mosconi et al., 2009*; *Elison et al., 2013*). A large-scale network, the default mode network, is also key since this network is one of the primary networks of the 'social brain' involved in high-level social-cognitive and social-communicative processing (e.g., mentalizing, joint attention) (*Lombardo et al., 2010a*; *Redcay et al., 2013*; *Van Overwalle, 2009*; *Schurz et al., 2014*; *Nummenmaa and Calder, 2009*; *Eggebrecht et al., 2017*; *Alcalá-López et al., 2018*; *Redcay and Schilbach, 2019*; *Schilbach et al., 2008*). We analyzed rsfMRI data in one of the largest and youngest samples of ASD to date and compare ASD to several non-ASD comparison groups – typically developing (TD) toddlers, toddlers with language or general developmental delay (LD/DD) and TD toddlers with an older ASD sibling (TD ASDSib). We also examined how connectivity differences may be better modeled by taking into account heterogeneity in early social visual engagement. Based on prior work theorizing altered connectivity between high-level social-cognitive networks in frontal cortex and posterior networks involved in sensation, perception and attention (*Courchesne and Pierce, 2005*), we hypothesized that there may be atypical and heterogeneous functional connectivity between higher-level social brain networks such as the default mode network (DMN) and posterior lower-level networks that are integral for visual perception and attention. We further hypothesized that GeoPref ASD toddlers may display the most extreme effects on functional connectivity. Finally, to test the utility of our ASD subtype for predicting individual differences in social-communication symptomatology, we predicted that robust relationships between functional connectivity and social-communication behavior would be apparent only within the GeoPref ASD subtype.

## Results

### Behavioral and developmental characteristics of the GeoPref ASD subtype

Our primary aim in this work was to test if early ASD subtypes defined on the basis of distinctions in early social engagement behavior would also differ at the level of macroscale neural circuit organization as measured with rsfMRI. However, because little prior work exists characterizing these subtypes, we first set out to detail how these subtypes are characterized both behaviorally and developmentally over the first 4 years of life. Using longitudinal data from 12 to 48 months of life, we assessed how the GeoPref and nonGeo ASD subtypes differ with respect to clinical behavioral trajectories. Every measure across the ADOS, Mullen Early Scales of Learning and the Vineland Adaptive Behavior subscales showed a significant effect of group, except for the Vineland Motor subscale. This indicates that there is an on-average difference between GeoPref and nonGeo ASD in this early developmental period manifesting with the GeoPref ASD subtype being more severely affected. In addition, the Mullen Receptive Language and Visual Reception subscales also showed evidence of an age*subtype interaction, which manifests as GeoPref ASD showing a much steeper downwards trajectory of severity compared to nonGeo ASD. These steeper downward trajectories in GeoPref ASD should not be interpreted as this subtype losing skills over development. Rather, since the scores analyzed here are normed T-scores, the correct interpretation is that as individuals get older, this subtype may be falling behind age-appropriate norms at a faster rate than nonGeo ASD.

No other measure showed evidence of this kind of age*subtype interaction (*Figure 1C*; *Figure 1— figure supplement 1* and *Supplementary file 1*). These insights generally extend prior work (*Pierce et al., 2011a*; *Pierce et al., 2016a*) by showing that the GeoPref ASD subtype distinction is indeed a subtype distinction with clinical importance extending beyond the domain of early social visual engagement. These results underscore that GeoPref ASD individuals are more severely affected across the ASD symptom dyad, as well as other language, cognitive, motor, and adaptive behavior domains. These data also suggest that receptive language and non-verbal cognitive skills may be particular domains where GeoPref ASD individuals may take on different developmental paths and progressively fall further behind typically-developing age norms compared to nonGeo ASD peers.

## Functional hypoconnectivity between the default mode network and visual or attention networks in ASD

To assess ASD differences in functional connectivity we modeled the rsfMRI data with two approaches – 1) an unstratified case-control model approach and 2) a stratified subtype model approach. These two modeling approaches can be empirically compared to discern whether subtyping by the GeoPref ASD or nonGeo ASD subtypes is an important distinction for explaining macroscale functional neural circuit organization. Using an unstratified case-control model comparing all ASD toddlers (n = 109) to toddlers from multiple non-ASD comparison groups (TD, TD ASDSib, LD/ DD; n = 55, n = 16, and n = 15 respectively), we identify three component pairs passing multiple comparisons correction at FDR q < 0.05. These effects feature reduced default mode network (DMN; IC10) connectivity in ASD with primary visual cortex (PVC; IC05: $F_{(3, 189)}=6.62$, p=2.79e-4, *partial $\eta^2$ = 0.099, Cohen's d > 0.52*), visual association circuitry located in occipito-temporal cortex (OTC; IC02: $F_{(3, 189)}=5.98$, p=6.41e-4, *partial $\eta^2$ = 0.088, Cohen's d > 0.51*) and the dorsal attention network (DAN; IC09: $F_{(3, 189)}=5.93$, p=6.87e-4, *partial $\eta^2$ = 0.087, Cohen's d > 0.40*) (*Figure 2*). All other comparisons between non-ASD comparison groups showed relatively small effect size differences, indicating that the ASD group is the primary reason for these larger reductions in connectivity. See *Supplementary file 2* for a full list of all statistics for these case-control comparisons. This modeling approach suggests that without taking into account heterogeneity in social engagement, ASD as a whole shows specific reductions in DMN connectivity with visual and attention circuits compared to TD, but also compared to LD/DD toddlers and TD siblings of ASD individuals.

## Marked functional hypoconnectivity between default mode and visual association circuitry in GeoPref ASD

Using the eye tracking subtypes from the GeoPref Test, we next recomputed stratified subtype models to test whether subtyping by social engagement heterogeneity provides a better explanation of differences in macroscale intrinsic functional neural circuit organization. DMN-OTC and DMN-PVC appear again in these stratified models as effects passing FDR q < 0.05, despite the fact that n = 31 ASD subjects from the original unstratified case-control model had to be dropped in this analysis since they lacked any eye tracking data to make the GeoPref or nonGeo ASD distinction. Descriptive standardized effect sizes for all pairwise comparisons of ASD subtypes and non-ASD comparison groups are shown in *Figure 2* and pairwise comparisons surviving FDR q < 0.05 correction are highlighted in *Figure 2* with black outlines around specific cells. See *Supplementary file 3* for statistics from pairwise group comparisons.

We then used the Akaike Information Criteria (AIC) to evaluate whether a stratified subtype model was better than the traditional unstratified case-control model. Models that result in a difference in AIC values (e.g., ΔAIC) less than or equal to 2, indicate that both models under comparison are largely similar to each other. However, models with ΔAIC ≥4 tend to show less support for the comparison model relative to the best model (i.e. the model with the lowest AIC) (*Burnham and Anderson, 2004*). Here we find that the stratified subtype model is the best performing model for DMN-OTC (IC10-IC02) connectivity, since it produces a lower AIC compared to the unstratified case-control model (*subtype model AIC = −51.32; case-control model AIC = −47.60*). The ΔAIC for DMN-OTC is 3.72, which is closest to the rule of 4 ≤ ΔAIC ≤7 suggested by *Burnham and Anderson (2004)* indicating substantially less support for the case-control model being a good contender against the subtype model. However, both DMN-PVC and DMN-DAN comparisons showed that the

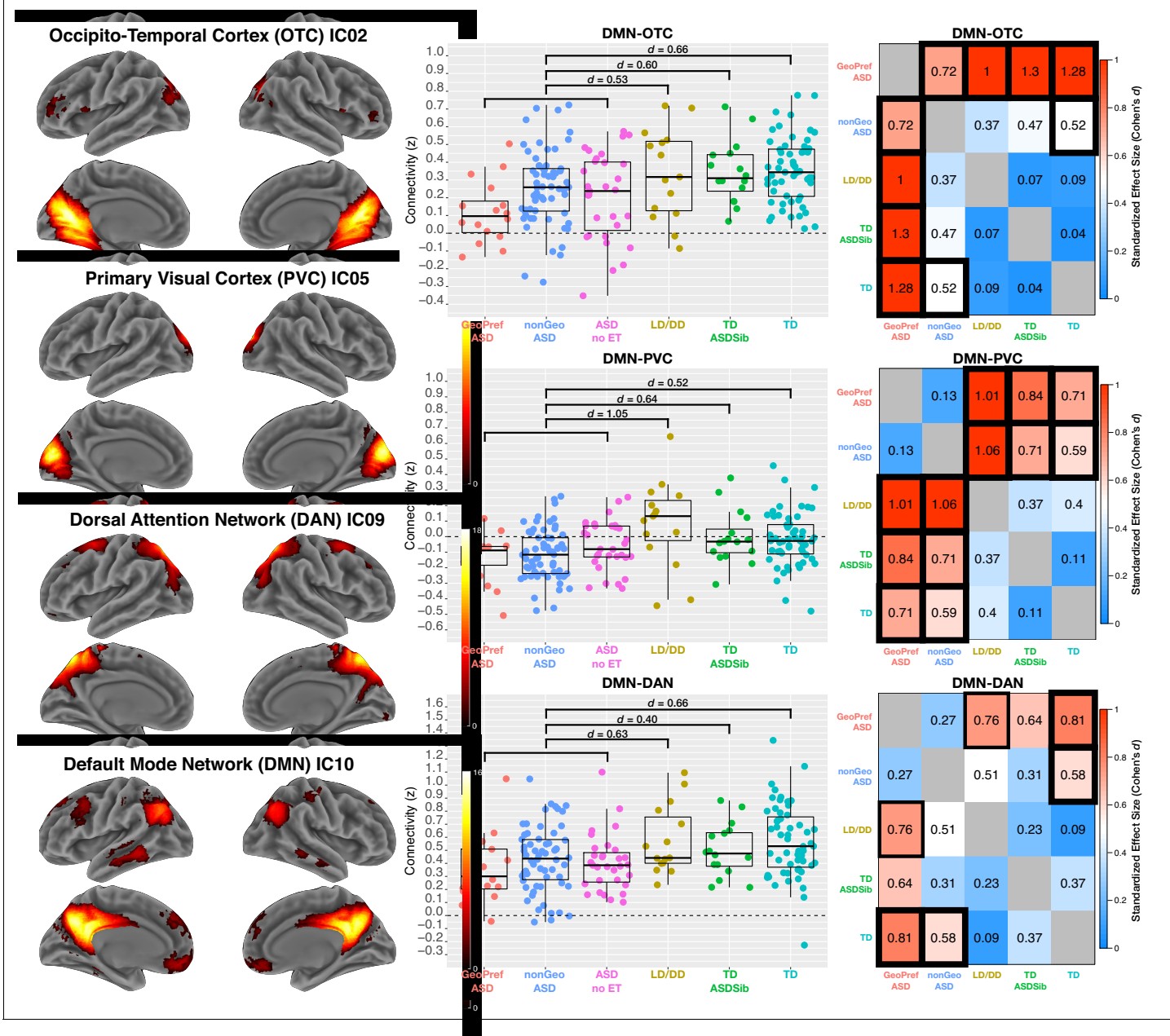

**Figure 2.** Functional hypoconnectivity between DMN, visual and attention networks in ASD. The left column shows surface renderings of ICA components of visual association areas in occipito-temporal cortex (OTC; IC02), primary visual cortex (PVC; IC05), the dorsal attention network (DAN; IC09) and the default mode network (DMN; IC10). The middle column shows scatter-boxplots for DMN-OTC, DMN-PVC, and DMN-DAN connectivity across GeoPref ASD (pink), nonGeo ASD (blue), ASD with no eye tracking data (ASD no ET; magenta), LD/DD (yellow), TD ASDSib (green), and TD (turquoise). Standardized effect sizes (Cohen's d) are reported in the plots for comparisons of all ASD individuals combined, compared to the other non-ASD comparison groups. The middle line of the boxplot represents the median. The box boundaries represent the interquartile range (IQR; Q1 = 25th percentile, Q3 = 75th percentile), while the whiskers indicate the a distance of 1.5*IQR. The right column uses heatmaps to show standardized effect sizes (Cohen's d) for all pairwise comparisons of groups with rsfMRI and eye tracking data used in the subtype model. Note that effect sizes are depicted as absolute values, and the directionality of the effects can be seen in the scatter-boxplots. Cells outlined in thick black lines are specific comparisons that survive FDR q < 0.05. Cells outlined in thinner black lines are comparisons that survive FDR q < 0.1.

The online version of this article includes the following figure supplement(s) for figure 2:

**Figure supplement 1.** Scatterplots of the relationship between functional connectivity by percentage fixation on the geometric stimulus.

**Figure supplement 2.** Simulation illustrating statistical power.

**Figure supplement 3.** Head motion effects.

stratified and unstratified models were largely similar to each other, with difference in AIC values consistently below 2 (DMN-PVC: *case-control model AIC = −89.54, subtype model AIC = −87.72*; ΔAIC = 1.82; DMN-DAN: *case-control model AIC = 19.20, subtype model AIC = 20.29*; ΔAIC = 1.09). In addition to ΔAIC, we also used 5-fold cross validation to compute mean absolute percentage error (MAPE). In this context, better models produce lower MAPE estimates. For DMN-OTC, MAPE was much lower for the subtype model (MAPE = 125.5452) compared to the case-control model (MAPE = 135.2241), indicating the subtype model reduced the absolute percentage of error by around 9.67%. However, for DMN-PVC and DMN-DAN, MAPE was lower for the case-control model (DMN-PVC subtype MAPE = 152.8099, case-control MAPE = 152.1738; DMN-DAN subtype MAPE = 156.0619, case-control MAPE = 152.8081), but in both cases the reduction in MAPE was <1%.

The subtype model can also be compared against models that ignore categorical subtype or diagnostic labels and instead model variability on continuous measures. Here we compared the subtype model to a transdiagnostic model using the continuous measure of fixation on the geometric stimulus from the GeoPref test as the primary predictor. The subtype model was unanimously a better explanatory model than a transdiagnostic model for each of the three component pairs, with ΔAIC >5 (DMN-OTC: subtype AIC = −51.32, transdiagnostic AIC = −46.13; ΔAIC = 5.19 DMN-PVC: subtype AIC = −87.72, transdiagnostic AIC = −74.63; ΔAIC = 13.09; DMN-DAN: subtype AIC = 20.29, transdiagnostic AIC = 25.42; ΔAIC = 5.13) (*Figure 2—figure supplement 1*).

These results generally suggest that heterogeneity in early social engagement is important for some, but not all, functional connectivity differences present in early ASD toddlers. This evidence supports the idea that in toddlerhood, ASD is characterized by functional hypoconnectivity between the DMN and visual circuits in PVC and OTC as well as attention circuitry (DAN). However, for the subtype of individuals with the most pronounced social engagement difficulties (GeoPref ASD), it is DMN-OTC connectivity that may be most important in distinguishing them from their other ASD counterparts.

## Subtype-specific association between DMN-OTC connectivity and social-communication difficulties

We next tested whether the two ASD subtypes would differ in the relationship between functional connectivity and social-communication difficulties. This test is critical for examining the question of whether the two ASD subtypes should indeed be considered as discrete types of ASD with different underlying neurobiological mechanisms relevant for the behavioral phenotype. Here we find evidence for subtype-specific associations for DMN-OTC connectivity. A strong negative correlation between DMN-OTC connectivity and ADOS social affect total is present in the GeoPref ASD subtype (*r* = −0.78, p=0.001, 95% CI = [−0.99,−0.35]), indicating that reductions in DMN-OTC connectivity are associated with more enhanced social-communication difficulties. No such relationship exists for nonGeo ASD (*r* = 0.06, p=0.64, 95% CI = [−0.20, 0.33]) (*Figure 3*). Correlation strength also significantly differed between the subtypes (z = 3.71, p=0.0001). In contrast, DMN-PVC showed no evidence of relationships in either subtype (GeoPref ASD *r* = 0.04, p=0.88, 95% CI = [−0.81, 0.75]; nonGeo ASD *r* = 0.22, p=0.12, 95% CI = [−0.10, 0.49]) or for differences in the strength of the relationship between subtypes (z = 0.59, p=0.55). DMN-DAN showed no evidence of a strong relationship in GeoPref ASD (*r* = −0.33, p=0.25, 95% CI = [−0.83, 0.58]). However, there was a significant, although weak positive relationship in nonGeo ASD (*r* = 0.29, p=0.04, 95% CI = [0.05, 0.49]). The differences in the strength of the relationship between subtypes was also weak, though significant (z = 2.14, p=0.03) (*Figure 3*).

## Discussion

Although social visual engagement difficulties are central early developmental features of ASD with high clinical and translational relevance, the neural bases behind these features are not well understood. Here we find robust evidence in early ASD development for atypical intrinsic functional brain organization involving circuitry relevant to social visual engagement. Central to our findings is the importance of early developing social brain circuitry, the default mode network (DMN) (*Lombardo et al., 2010a*; *Kennedy and Adolphs, 2012*; *Kennedy and Courchesne, 2008a*; *Kennedy et al., 2006*; *Padmanabhan et al., 2017*; *Lombardo et al., 2011*; *Lombardo et al.,*

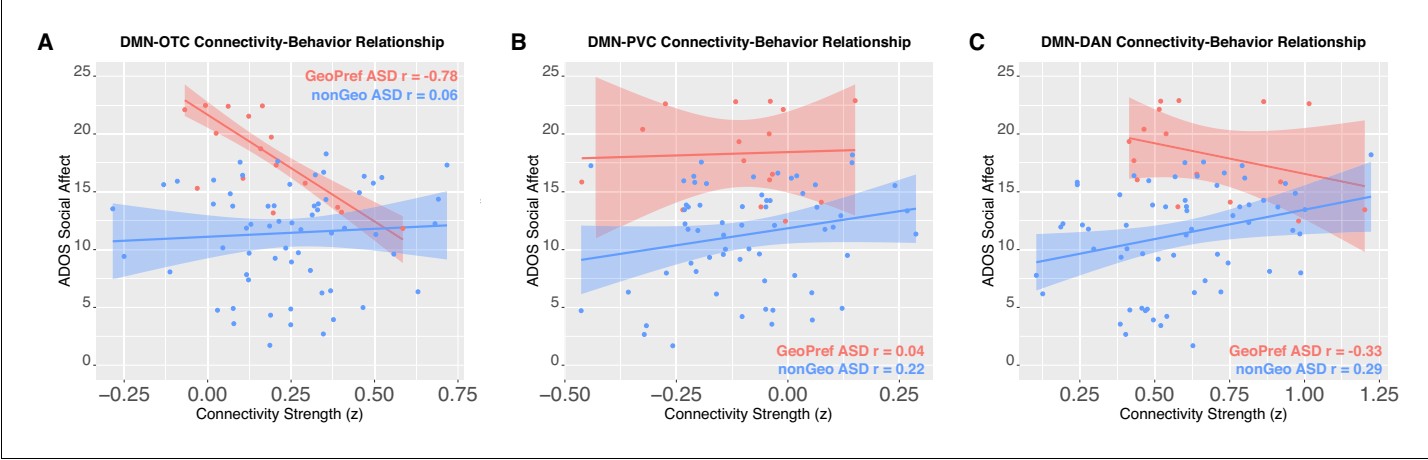

**Figure 3.** Connectivity-social-communication relationships. This figure shows functional connectivity-social-communication relationships for DMN-OTC (Panel **A**), DMN-PVC (Panel **B**), and DMN-DAN (Panel **C**). The relationship for GeoPref ASD is shown in pink, while nonGeo ASD is shown in blue. Both ADOS social affect and connectivity scores shown in the plot are covariate adjusted scores (taking into account age) from the robust regression model evaluating the relationship.

*2010b*), and its interactions with more basic neural circuitry involved in visual or attention processes. The DMN has been extensively studied in older ASD patients and has been found to be atypical in multiple studies using resting state and task-based social paradigms (*Padmanabhan et al., 2017*; *Lombardo et al., 2010b*; *Di Martino et al., 2014*; *Kennedy and Courchesne, 2008b*; *Hong et al., 2019*; *Lai et al., 2019*; *Holiga et al., 2019*). However, a novel addition to this literature is our work here showing that in very early development there is functional hypoconnectivity between DMN and primary visual cortex (PVC) as well as visual association circuitry in occipito-temporal cortex (OTC). Here we find that at this point in early development normative groups (e.g., TD) show robust non-zero DMN-OTC connectivity. In contrast, in older typically-developing samples, similar types of robust functional connectivity relationships between these networks are not extensively reported in literature, as DMN is typically highly segregated and embedded at an opposite pole within functional connectivity gradients from sensory cortices (*Smith et al., 2015*; *Margulies et al., 2016*). These effects along with DMN-dorsal attention network (DAN) hypoconnectivity show evidence for a case-control difference compared to TD and other non-ASD comparison toddler age groups, and these effects are prominent across both ASD cohorts with and without eye tracking data. This finding suggests that in ASD within the first 4 years of life the developing 'social brain' default mode network is on-average functionally disconnected from visual cortices and the dorsal attention network. This finding is important with respect to why social engagement difficulties are a hallmark symptom in early ASD development and shared across most ASD toddlers.

In contrast to generalized case-control differences, we also observed key evidence that early heterogeneity in social visual engagement is important for explaining functional hypoconnectivity between DMN and OTC. The GeoPref ASD subtype is a primary factor behind DMN-OTC functional hypoconnectivity in ASD and this subtype distinction is a better explanatory model than a traditional case-control model or other transdiagnostic models. While GeoPref ASD is behaviorally more severe on a range of different domains, this functional connectivity difference is specific to GeoPref ASD. Other non-ASD toddlers with general developmental or language delay do not show such effects. Furthermore, the magnitude of DMN-OTC functional hypoconnectivity specific to GeoPref ASD was strongly associated with individual-level variability in social-communication difficulties measured on an ASD 'gold standard' diagnostic instrument (i.e. ADOS). While DMN-PVC and DMN-DAN connectivity reductions are also prominent in GeoPref ASD toddlers, it could be DMN-OTC connectivity that has the largest implications for the different clinical presentation specific to this subtype. Early functional hypoconnectivity between DMN-OTC circuitry could have important differential impact for this subtype with regards to further developmental functional specialization of the social brain for social cognitive and social-communicative functions. In contrast, the less affected DMN-OTC circuitry in nonGeo ASD individuals could be an indication of the plasticity in this circuitry as a result of

relatively more enhanced social experience in this subtype. Given that many early interventions heavily focus on enhancing early social experience (*Bacon et al., 2014*; *Dawson et al., 2010*; *Kasari et al., 2006*; *Pickles et al., 2016*; *Dawson, 2008*), DMN-OTC connectivity may be a key target for better understanding why individuals heavily vary in response to early intervention treatment.

It is unknown what neurobiological processes underlie these differences in DMN connectivity with visual and attention circuitry. Recent evidence from two twin studies suggests that variability in early social visual engagement is underpinned by genetic factors potentially linked to heritable common genetic variants (*Constantino et al., 2017*; *Kennedy et al., 2017*). How such potential genetic starting points lead to further pathophysiology up to the behavioral phenotype is still a mystery. However, it is clear from recent work that neural pathophysiology in ASD likely begins at the earliest prenatal periods of development (*Courchesne et al., 2011*; *Stoner et al., 2014*; *Parikshak et al., 2013*; *Willsey et al., 2013*; *Marchetto et al., 2017*; *Parikshak et al., 2016*; *Courchesne et al., 2019*). This prenatal pathophysiology is diverse but includes increased cell proliferation, aberrant cell migration, and downregulated early axonogenesis during early prenatal periods, whereas starting in later prenatal development and continuing throughout the first years of life there is also downregulation of processes typically important for synapse development (*Courchesne et al., 2011*; *Stoner et al., 2014*; *Parikshak et al., 2013*; *Willsey et al., 2013*; *Marchetto et al., 2017*; *Parikshak et al., 2016*; *Courchesne et al., 2019*). Microglia activation and upregulation of immune/inflammation and protein synthesis (i.e. translation initiation) processes are prominent in ASD throughout life and could have further potential impact on cell signaling relevant to neuronal, glial, and synaptic development (*Parikshak et al., 2016*; *Lombardo et al., 2017*; *Lombardo et al., 2018b*; *Pramparo et al., 2015a*; *Gupta et al., 2014*; *Voineagu et al., 2011*; *Chow et al., 2012*; *Morgan et al., 2010*; *Takano, 2015*). Collectively, postmortem, genomic and genetic, early-age neuroimaging, animal model and patient induced pluripotent stem cell (iPSC) model studies indicate prenatal and postnatal ASD cortex displays an abnormal overabundance of small and possibly poorly differentiated neurons with undergrown axons (*Solso et al., 2016*), focal laminar and migration defects, neuroinflammation and aberrant synapse formation and function (*Courchesne et al., 2019*). It has long been theorized that such early abnormalities could cause disconnection of higher-order frontal social and communication networks from lower-level posterior perceptual networks and thereby impair attention to and integration of relevant social, emotional and communicative events during development in ASD (*Courchesne and Pierce, 2005*). While these early developmental abnormalities are numerous and diverse, a key goal for future work will be to determine which processes, or collection of processes working together, lead to the early emergence of ASD and different social engagement subtypes. One compelling hypothesis is that prenatal pathophysiology could cause a cascade in early development leading towards the ultimate initiation of atypical social engagement behaviors. Aggregation of common genetic risk associated with atypical social visual engagement (*Constantino et al., 2017*; *Kennedy et al., 2017*) may be relevant to atypical prenatal development processes (*Courchesne et al., 2019*). These atypical prenatal processes could trigger early behavioral adaptation responses that manifest as different ways to explore and sample the social environment in early development (*Johnson et al., 2015*; *Johnson, 2017*). Different social visual engagement behavior could then lead to different experience-dependent brain development in different ASD toddlers. The end result of this chain could be the emergence of behavioral and neural subtypes but also with some shared aspects of neural abnormalities that are specific to ASD. It will be crucial for future work to examine what set of genetic risk variants and early prenatal pathophysiology may underpin variation in early social visual engagement in ASD and how such mechanisms may lead to different experience-dependent development and neural circuitry. Work utilizing iPSC modeling to recapitulate some of the early prenatal neurobiological processes occurring in such individuals seems highly appropriate for beginning to answer such questions (*Marchetto et al., 2017*).

The current work may also be important for explaining how and why early developmental functional specialization of social brain circuitry occurs in typical development, and how it may be atypical in some individuals with ASD. Reduced early social engagement behaviors that are adaptations to atypical prenatal starting points could potentially lead to the construction and long-term maintenance of atypical early developmental environmental niches for an individual (*Johnson et al., 2015*; *Johnson, 2017*). An environmental niche constructed and maintained from largely sampling the non-social instead of the social world around an infant could have a large impact on how postnatal neural

circuits are formed. It is well known that neural circuits are sculpted by experience, particularly in the early years of life when biological processes such as synaptogenesis, axon expansion, and cell-growth are normally at their peak (*Greenough et al., 1987*; *Holtmaat and Svoboda, 2009*; *Hutten-locher, 2002*; *Kang et al., 2011*). Selective biases to the type of information sampled from the environment in GeoPref ASD may provide the wrong type of input to facilitate social cognitive and social-communicative functional specialization of circuits such as the DMN. Circuits such as the DMN undergo protracted developmental periods of interactive specialization (*Johnson, 2011*), whereby specialization is achieved through interactions between circuits. Our work here suggests that functional interaction between visual and social brain circuits may be crucial to early social behavior in this ASD subtype, and may have further impact for aiding the developing specialization of function within social brain circuitry.

In order to put this work into context, there are a variety of strengths, caveats, and limitations to underscore. Methodologically, the findings are bolstered by being one of the largest sample sizes to date of rsfMRI data in very young ASD toddlers. The early age range of our sample stands out with respect to most of the existing rsfMRI evidence in the literature, since a large majority of studies examine much older individuals. An interpretational caveat with studies of older individuals is that it is unclear which effects are due to core pathophysiology relevant to ASD and which emerged at much earlier developmental time-points versus effects that could be explained as adaptation or other compensatory mechanisms that occur later in life and are not necessarily linked to core early neurodevelopmental mechanisms of relevance to ASD. While adaptation and compensatory effects are still likely even at very early ages, the advantage of the early age range in this sample (relative to studies on older individuals) is the enhanced ability to interpret the results with regards to possible core mechanisms that are at work at ages when the earliest behavioral symptoms relevant to ASD diagnosis manifest.

The findings are also bolstered by a methodological strength of making comparisons against other non-ASD comparison groups, such as LD/DD and TD ASDSib. Such non-ASD and non-TD comparison groups are relatively rare compared to most neuroimaging studies in the current literature. Comparisons between such groups allow for added inferences about specificity of effects to ASD or its subtypes and also help to show that these effects are not simply due to being clinically more severe across a range of domains. The sample ascertainment strategy based on early population-screening is another considerable strength that should result in higher generalizability compared with other studies that study a smaller subsection of the population, such as baby sibling studies. In fact, data generated using this unique sample ascertainment strategy revealed that functional connectivity within the examined networks including between the DMN and OTC did not significantly differ between unaffected siblings and typically developing toddlers. Interpretation of this finding warrants further investigation but could suggest that the development of early connectivity between social and visual attention brain networks could be key to whether or not an infant eventually manifests symptoms of ASD.

Finally, our study goes beyond most work using case-control designs and identifies heterogeneity in a relatively infrequent subtype that comprises around 20% of the early ASD population. In this study, the sample size of the GeoPref ASD subtype was relatively small (n = 16). While small sample size can be problematic for reasons of suboptimal coverage of the ASD spectrum in case-control comparisons (*Lombardo et al., 2019*), our design here confronts this issue head first by parsing some aspect of the heterogeneity in ASD, rather than simply being a small sample size case-control comparison. However, because of the imbalance in sample sizes between GeoPref and nonGeo ASD subtypes, future work is necessary where the sample sizes are more balanced. This will require much effort to enrich samples with the GeoPref ASD subtype, given that it is present in around 20% of the early diagnosed ASD population. Second, while small sample size can be problematic for statistical power reasons, we have shown what are the minimal effect sizes at such sample sizes for adequate statistical power (*Figure 2—figure supplement 2*), and our estimated effect sizes are well above such minimal effects sizes. The brain-behavior correlation estimates are also supplemented by the reporting of bootstrap confidence intervals. Since small sample sizes can result in inflated estimates of correlation, the bootstrap confidence intervals allow for reporting of the distribution of sample correlation estimates that could have been observed.

Finally, it is important to underscore that the GeoPref ASD subtype is based on one relatively quick and simple early eye tracking test. This test picks up about 20% of all early diagnosed ASD

individuals as within this subtype. It could be that integration of a larger battery of eye tracking measures and tests may help enhance sensitivity and detect a larger percentage of individuals that may be part of this subtype (*Moore et al., 2018*). Furthermore, use of eye tracking measures that go beyond static stimuli and use interaction-based paradigms (*Redcay and Schilbach, 2019*; *Schilbach et al., 2013*) may further enhance sensitivity. All of these refinements could potentially aid in the identification of early biological bases behind different ASD subtypes.

In conclusion, we identified that functional hypoconnectivity between 'social brain' circuitry, the DMN, and low-level visual networks is highly important in the early development of ASD. While DMN-PVC functional connectivity is reduced on-average across ASD toddlers, DMN-OTC functional connectivity is heavily reduced in the GeoPref ASD subtype. Individual-level variation in DMN-OTC functional hypoconnectivity is associated with the degree of social-communication difficulties, but only within the GeoPref ASD subtype. This subtype can be identified with high levels of precision via a simple eye tracking test of social or non-social visual preferences in very early development (*Pierce et al., 2011a*). Early social-visual functional hypoconnectivity is a key underlying neurobiological feature describing GeoPref ASD and may be critical for future social-communicative development. Thus, we theorize that a neurobiologically driven bias (particularly from prenatal development) leading towards neglect for the social world in early development, if left untreated, would significantly diminish opportunities for social learning and for bootstrapping experience-dependent change within developing neural circuits. This, we suggest, may limit developmental functional specialization within the social brain and lead towards more permanent long-term social, communicative, and cognitive difficulties. Therefore, it is an important goal to hone in on early-age molecular biomarkers that personalize risk, since very early identification may have larger clinical and translational benefits under such contexts. Early detection and interventions that successfully ameliorate early atypical functional connectivity between social brain and visual circuitry might improve social development and outcome for ASD individuals. Despite the narrow view of some institutions (USPSTF) (*Siu et al., 2016*), early risk detection is an absolute necessity (*Pierce et al., 2016b*), as is research to devise effective early interventions tailored to specific ASD individuals.

## Materials and methods

### Participants

This study was approved by the Institutional Review Board at University of California, San Diego (UCSD Human Research Protection Program protocols 091539, 081722, or 110049). Parents provided written informed consent according to the Declaration of Helsinki and were paid for their participation. Identical to the approach used in our earlier studies (*Pierce et al., 2011a*; *Pierce et al., 2016a*; *Lombardo et al., 2015*; *Pramparo et al., 2015a*; *Pramparo et al., 2015b*), toddlers were recruited through two mechanisms: community referrals (e.g., website) or a general population-based screening method called the 1 Year Well-Baby Check-Up Approach (*Pierce et al., 2011b*) that allowed for the prospective study of ASD beginning at 12 months based on a toddler's failure of the CSBS-DP Infant-Toddler Checklist (*Wetherby and Prizant, 2002*; *Wetherby et al., 2008*). All toddlers were tracked from an intake assessment age of 12–24 months and followed roughly every 12 months until 3–4 years of age. All toddlers, including normal control subjects, participated in a series of tests collected longitudinally across all visits, including the Autism Diagnostic Observation Schedule (ADOS; Module T, 1, or 2) (*Lord et al., 2000*), the Mullen Scales of Early Learning (*Mullen, 1995*), and the Vineland Adaptive Behavior Scales (*Sparrow et al., 1984*). All testing occurred at the University of California, San Diego Autism Center of Excellence (ACE).

A total of n = 195 toddlers aged 12 to 48 months were scanned with rsfMRI during natural sleep for the current study. Of the total n = 195, n = 109 were ASD toddlers and this ASD group could be further split into n = 16 GeoPref ASD toddlers (11 male, five female), n = 62 nonGeo ASD toddlers (49 male, 13 female), and a further n = 31 ASD toddlers (27 male, four female) that could not be stratified in either GeoPref or nonGeo ASD subtypes because they lacked eye tracking data needed for such stratification. The GeoPref ASD subtype is defined as toddlers who spent 69% or more of their time fixated on the dynamic geometric stimulus in the GeoPref eye tracking test (*Pierce et al., 2011a*; *Pierce et al., 2016a*). The nonGeo ASD subtype represents individuals that spent less than 69% of time fixated on the dynamic geometric stimulus. The 69% threshold for the GeoPref test has

been shown in past studies to replicably isolate the GeoPref ASD subtype and maximize specificity with respect to many other contrast groups (*Pierce et al., 2011a*; *Pierce et al., 2016a*). Several toddlers from multiple non-ASD groups were examined as contrast groups to the ASD sample. These toddlers consisted of n = 55 typically-developing control toddlers (TD, 37 male, 18 female), n = 15 with language or globally developmental delay (LD/DD, 10 male, five female), and n = 16 TD toddlers who were younger siblings of children already diagnosed with ASD (TD ASDSib, eight male, eight female). See *Table 1* for characteristics such as age at scanning and age at eye tracking for all groups. Amongst the total n = 195 toddlers with rsfMRI data, an ANOVA on age at scanning found no group-differences across the groups ($F(5, 189)$=0.89, p=0.48). A chi-square analysis across the total n = 195 identified a subtle trend-level difference in the proportion of males and females distributed across groups, with more even proportions of males and females in non-ASD comparison groups than the ASD groups ($\chi^2(5)$=9.88, p=0.07). In all subsequent connectivity analyses, we statistically controlled for sex and age at scanning as covariates.

Because the n = 16 within the GeoPref ASD subtype is a relatively small sample size, we ran a statistical power analysis simulation to identify the minimum effect size needed to detect reject the null hypothesis with an alpha of 0.05% and 80% power at the current sample sizes (e.g., n = 16 GeoPref ASD vs n = 55 TD). In this simulation, we generated data from two populations (hypothetically GeoPref ASD and TD), with the minimal effect size difference between the populations that achieves 80% power at these sample sizes. The population size of each group was set to n = 10,000,000. We then ran 100,000 simulated experiments whereby we randomly sampled without replacement n = 16 (e.g., GeoPref ASD) and n = 55 (e.g., TD) from each population, and then computed effect size and ran a hypothesis test (e.g., independent samples t-test) with the alpha set to 0.05. In *Figure 2—figure supplement 2*, we illustrate how variable sample effect size estimates can be given this context of n = 16 vs n = 55 (e.g., sample effect size estimates range from d < 0 to d > 1.5). Power is empirically shown here (see red histogram in *Figure 2—figure supplement 2*) since exactly 80,000 of the 100,000 (80%) experiments rejected the null hypothesis at alpha = 0.05. The minimum effect size to achieve 80% power with these sample sizes is d = 0.80751.

## Eye tracking paradigm

Eye-tracking data from this study has already been reported in two prior studies (*Pierce et al., 2011a*; *Pierce et al., 2016a*). Briefly recapping the GeoPref eye tracking test, toddlers were seated on their parent's lap 60 cm in front of the Tobii T120 eye tracking monitor and were presented a movie consisting of dynamic geometric or social stimuli on either side of the screen. The dynamic geometric stimulus was produced from recordings of animated screen saver programs. The dynamic social stimulus was produced from a series of short sequences of children doing yoga, which included images of children moving in a dramatic manner (e. g., waving arms and appearing as if dancing). These clips were used with permission from the commercially available video Yoga Kids 3 (2003 Gaiam Americas, Inc; https://www.gaiam.com) and specific permissions were given to use the clips for research purposes in published scholarly work. Audio information was discarded. The final stimulus was composed of 2 rectangular areas of interest (AOIs) horizontally distributed containing either the geometric or social stimulus and were changed in a simultaneous, time-linked fashion. The

**Table 1.** Descriptive statistics across all groups for age at rsfMRI scan, sex, and head motion measurements.

| | N (M, F) | Mean age at eye tracking in months (SD) | Mean age at rsfMRI scan in months (SD) | Age range at rsfMRI scan in months | Mean framewise displacement (SD) |
|---|---|---|---|---|---|
| ASD no ET (rsfMRI data only; no eye tracking data) | 31 (27,4) | - | 29.69 (8.88) | 13.21–43.63 | 0.09 (0.08) |
| GeoPref ASD | 16 (11,5) | 28.37 (7.77) | 29.92 (8.71) | 14.16–43.79 | 0.06 (0.02) |
| nonGeo ASD | 62 (49,13) | 26.30 (8.35) | 29.37 (8.35) | 12.35–44.05 | 0.09 (0.12) |
| LD/DD | 15 (10,5) | 19.36 (4.15) | 25.12 (7.97) | 13.37–39.75 | 0.10 (0.05) |
| TD ASDSib | 16 (8,8) | 19.79 (6.20) | 26.74 (9.38) | 12.52–44.09 | 0.08 (0.04) |
| TD | 55 (37,18) | 23.07 (9.07) | 29.61 (10.14) | 13.17–47.93 | 0.10 (0.11) |

side (left/right) of presentation of geometric or social stimuli were randomly assigned across subject and diagnosis and percent fixation within each AOI calculated. The final movie contained a total of 28 scenes with single-scene duration varying from 2 to 4 s for a total presentation time of 60 s at 24 frames per second.

## Longitudinal clinical behavioral data

To examine behavioral trajectories on other clinical measures (i.e. ADOS, Mullen, Vineland) we utilized the largest sample of GeoPref ASD toddlers available to boost power beyond the smaller sample that also had rsfMRI data available. An additional 44 GeoPref ASD toddlers who had valid eye tracking and clinical data (but not rsfMRI) were included with the n = 16 GeoPref ASD toddlers with eye tracking and rsfMRI data. Thus, a total of n = 60 GeoPref ASD and n = 62 nonGeo ASD toddlers were analyzed for differences in clinical behavioral trajectories, as illustrated in *Figure 1C* and *Figure 1—figure supplement 1*. We used linear mixed-effect modeling analyses (modeling random slopes and intercepts) to model within-individual trajectories and group-level trajectories. These analyses were implemented on z-scored data with the *lme* function contained within the *nlme* library in R. The dependent variables were either ADOS, Mullen, or Vineland subscale scores, while the independent variables modeled were always age, group, and the age*group interaction. A comparison of linear versus quadratic models indicated that a linear model provided a better fit (linear model, AIC = −158.09; quadratic model, AIC = −148.64) and thus, the linear model was used.

## fMRI data acquisition

Imaging data were collected on a 1.5 Tesla General Electric MRI scanner during natural sleep at night; no sedation was used. High-resolution T1-weighted anatomical scans were collected for warping fMRI data into standard atlas space. Blood oxygenation level-dependent (BOLD) signal was measured across the whole brain with echoplanar imaging (TE = 30 ms, TR = 2500 ms, flip angle = 90°, bandwidth = 70 kHz, field of view = 25.6 cm, in-plane resolution = 4×4 mm, slice thickness = 4 mm, 31 slices). The resting state session lasted 6 min and 25 s resulting in 154 total whole brain volumes. Within the preprocessing we discarded the first four volumes to allow for T2-stabilization effects, leaving a total of 150 volumes for the final analysis.

## fMRI data preprocessing

Preprocessing of the resting state data was split into two components; core preprocessing and denoising. Core preprocessing was implemented with AFNI (http://afni.nimh.nih.gov/) using the tool speedypp.py (http://bit.ly/23u2vZp) (*Kundu et al., 2012*). This core preprocessing pipeline included the following steps: (i) slice acquisition correction using heptic (7th order) Lagrange polynomial interpolation; (ii) rigid-body head movement correction to the first frame of data, using quintic (5th order) polynomial interpolation to estimate the realignment parameters (3 displacements and three rotations); (iii) obliquity transform to the structural image; (iv) affine co-registration to the skull-stripped structural image using a gray matter mask; (v) nonlinear warping to MNI space (MNI152 template) with AFNI *3dQwarp*; (v) spatial smoothing (6 mm FWHM); and (vi) a within-run intensity normalization to a whole-brain median of 1000. Core preprocessing was followed by denoising steps to further remove motion-related and other artifacts. Denoising steps included: (vii) wavelet time series despiking ('wavelet denoising'); (viii) confound signal regression including the six motion parameters estimated in (ii), their first order temporal derivatives, and ventricular cerebrospinal fluid (CSF) signal (referred to as 13-parameter regression). The wavelet denoising method has been shown to mitigate substantial spatial and temporal heterogeneity in motion-related artifact that manifests linearly or non-linearly and can do so without the need for data scrubbing (*Patel et al., 2014*). Wavelet denoising is implemented with the Brain Wavelet toolbox (http://www.brainwavelet.org). The 13-parameter regression of motion and CSF signals was achieved using AFNI *3dBandpass* with the *–ort* argument. To further characterize and describe motion and its impact on the data, we computed framewise displacement and DVARS (*Power et al., 2012*). Examples of how denoising impacts high and low motion subjects can be found in the *Figure 2—figure supplement 3*. Between-group comparisons showed that all groups were similar with respect to head motion as measured by mean framewise displacement (FD) ($F(5,189)$ = 0.67, p=0.64) with all groups showing on-average less than 0.12 mm motion (see *Table 1*). Furthermore, mean DVARS measurements were similar across all groups

before ($F(5,189)$ = 0.28, p=0.92) and after denoising ($F(5,189)$ = 0.57, p=0.72). Both of these results indicate that motion does not asymmetrically affect one group more than the others.

## Connectivity analyses

To assess functional connectivity between neural circuits we utilized the unsupervised data-driven method of independent component analysis (ICA) to conduct a group-ICA and then used dual regression to back-project spatial maps and individual time series for each component and subject. Both group-ICA and dual regression were implemented with FSL's MELODIC and Dual Regression tools (www.fmrib.ox.ac.uk/fsl). For group-ICA, the dimensionality estimate was fixed to a pre-specified dimensionality of 30 components, as in most cases with low-dimensional ICA, the number of meaningful components can be anywhere from 10 to 30 (*Smith et al., 2013*). Higher-dimensional solutions were not sought, although they may be important in future work, particularly with regard to further fractionating known networks like the DMN into subsystems (*Kernbach et al., 2018*). Given our a priori hypotheses regarding specific processes such as social cognition, attention, salience, visual perceptive, affective, and reward processes, we examined corresponding known networks that are involved in such processes – namely, the default mode, dorsal attention, salience, several visual networks, and amygdala/striatum-centered networks.

Time courses for each subject and each component were used to model between-component connectivity. This was achieved by constructing a partial correlation matrix using Tikhonov-regularization (i.e. ridge regression, rho = 1) as implemented within the *nets_netmats.m* function in the FSLNets MATLAB toolbox (https://fsl.fmrib.ox.ac.uk/fsl/fslwiki/FSLNets). The aim of utilizing partial correlations was to estimate direct connection strengths in a more accurate manner than can be achieved with full correlations, which allow more for indirect connections to influence connectivity strength (*Smith et al., 2013*; *Marrelec et al., 2006*; *Smith et al., 2011*). Partial correlations were then converted into Z-statistics using Fisher's transformation for further statistical analyses and the lower diagonal of each subject's partial correlation matrix was extracted.

General linear models (GLM) were utilized to test for between-group differences in partial correlations – 1) an unstratified case-control model and 2) a stratified subtype model. Both GLM models were implemented with the lm function in R. In these GLMs the partial correlation for a particular component pair was the dependent variable. For independent variables, in all GLMs we used sex and age at scanning as covariates of no interest. The primary independent variable of interest was the group variable. Under the case-control model, group was composed of ASD and non-ASD labels. ASD was the sole label for all ASD subjects and no stratification by eye tracking data was done. The controls in this model were the other non-ASD comparison groups – TD, LD/DD and TD ASDSib. These groups were treated as other separate groups in the model rather than being collapsed into one 'control' group. In the stratified subtype model, group utilized the ASD subtype labels identified through the eye tracking data – either the GeoPref and nonGeo ASD subtypes. Group labels for each non-ASD comparison group were also used in this group variable, as was the case for the case-control model. These models were computed for each component pair analyzed and results were thresholded at FDR q < 0.05 for multiple comparisons correction. If any component pairs survived FDR correction, these components were further followed-up with analyses on each pairwise group-comparison and with FDR correction at q < 0.05. These pairwise group-comparisons were analyses using non-parametric permutation tests (10,000 permutations), in order to estimate p-values for each pairwise comparison in a manner that is robust to distributional assumptions. Permutation p-values were then used to compute FDR and only comparisons passing FDR q < 0.05 were deemed significant. Effect sizes from all pairwise group comparisons were estimated as standardized effect size (Cohen's d) using the cohen.d function in the effsize library in R.

To compare the case-control model to the subtype model, we utilized the Akaike Information Criteria (AIC) statistic. AIC was chosen as the model comparison criterion since it is optimal, compared to other criteria such as BIC, under contexts similar to ASD where the true reality is one of complexity or 'tapering effects' (*Burnham and Anderson, 2004*). In the context of comparing models, typically the model with the lowest AIC value is considered the best model. However, to facilitate interpretation regarding whether there was strong support for either of the models being compared, we computed the difference in AIC, whereby $\Delta AIC_i = AIC_i - AIC_{min}$. $AIC_{min}$ is the AIC for the model with the lowest AIC value and $AIC_i$ is the AIC for the i-th model being compared. Burham and Anderson (*Burnham and Anderson, 2004*) note that $\Delta AIC_i \leq 2$ indicates that the $AIC_i$ model has

strong support for being equally as good as the $AIC_{min}$ model. When $4 \leq \Delta AIC_i \leq 7$, this indicates considerably less support for the $AIC_i$ model compared to the $AIC_{min}$ model. To further support model selection, we also implemented 5-fold cross validation to compute mean absolute percentage error (MAPE) on held-out unseen data. MAPE is computed as MAPE = mean(abs(($A_i - P_i$)/$A_i$)*100), where $A_i$ is actual test data point i, $P_i$ is predicted test data point i, and abs refers to the absolute value. The model with the lowest MAPE is the best model to choose, and we report these results alongside $\Delta AIC_i$. MAPE is computed for each cross validation fold, and then the average is reported across folds. The difference in MAPE values also allows for interpretability with regards to how much of a reduction in percentage error is gained by the better model.

We also used AIC and $\Delta AIC_i$ to compare the subtype model to a continuous transdiagnostic model. These models utilized all data from all groups with both fMRI and eye tracking data. The continuous transdiagnostic model utilized the continuous measure of percent fixation on the geometric stimulus and without any variable indicating group membership.

## Connectivity-Behavior relationships

To test for relationships between connectivity and social-communication difficulties on the ADOS, we computed robust regression partial correlations between connectivity and ADOS social affect total scores, while covarying for age at scanning. These correlations were computed using the robust regression MATLAB toolbox (https://github.com/canlab/RobustToolbox; *Wager et al., 2005*). To test for difference between-groups in the strength of such correlations we used the *paired.r* function in the *psych* R library to compute Z-statistics and p-values. We also performed bootstrapping to compute 95% confidence intervals around sample correlation estimates, using 100,000 bootstrap resamples. This analysis was done to allow for reporting of the distribution of sample correlation estimates that could have been observed.

## Data and code availability

Tidy data and analysis code are available at https://github.com/mvlombardo/geoprefrsfmri (*Lombardo, 2019*; copy archived at https://github.com/elifesciences-publications/geoprefrsfmri).

## Acknowledgements

This research was supported by grants NIMH R01-MH080134 (EC, KP), NFAR grant (KP), NIMH Autism Center of Excellence grant P50-MH081755 (EC, KP), NIDCD R01-DC016385 (EC, KP), CDMRP AR130409 (EC), and an ERC Starting Grant (ERC-2017-STG; 755816) to MVL and fellowships from Jesus College, Cambridge and the British Academy to MVL. We thank Richard Znamirowski, Clelia Ahrens-Barbeau, Stephanie Solso, Kathleen Campbell, Maisi Mayo, and Julia Young for help with data collection, Stuart Spendlove and Melanie Weinfeld for assistance with clinical characterization of subjects.

## Additional information

### Competing interests

Karen Pierce: An invention disclosure form was filed by KP with the University of California, San Diego, on March 5, 2010. The other authors declare that no competing interests exist.

### Funding

| Funder | Grant reference number | Author |
| --- | --- | --- |
| H2020 European Research Council | 755816 | Michael V Lombardo |
| National Institute of Mental Health | R01-MH080134 | Eric Courchesne Karen Pierce |
| National Institute of Mental Health | P50-MH081755 | Eric Courchesne Karen Pierce |

| National Institute on Deafness and Other Communication Disorders | R01-DC016385 | Michael V Lombardo Eric Courchesne Karen Pierce |
| --- | --- | --- |
| CDMRP | AR130409 | Eric Courchesne |
| Jesus College, University of Cambridge | Fellowship | Michael V Lombardo |
| British Academy | Fellowship | Michael V Lombardo |

The funders had no role in study design, data collection and interpretation, or the decision to submit the work for publication.

## Author contributions

Michael V Lombardo, Conceptualization, Data curation, Software, Formal analysis, Funding acquisition, Visualization, Methodology, Writing—original draft, Writing—review and editing; Lisa Eyler, Investigation, Methodology, Writing—review and editing, Data collection; Adrienne Moore, Cynthia Carter Barnes, Debra Cha, Writing—review and editing, Data collection; Michael Datko, Methodology, Writing—review and editing; Eric Courchesne, Karen Pierce, Conceptualization, Supervision, Funding acquisition, Methodology, Writing—original draft, Project administration, Writing—review and editing, Data collection

## Author ORCIDs

Michael V Lombardo (iD) https://orcid.org/0000-0001-6780-8619

## Ethics

Human subjects: This study was approved by the Institutional Review Board at University of California, San Diego (UCSD Human Research Protection Program protocols 091539, 081722, or 110049). Parents provided written informed consent according to the Declaration of Helsinki and were paid for their participation.

## Decision letter and Author response

Decision letter https://doi.org/10.7554/eLife.47427.sa1
Author response https://doi.org/10.7554/eLife.47427.sa2

# Additional files

## Supplementary files

• Supplementary file 1. Statistics for longitudinal behavior analyses. This file shows statistics for linear mixed effect models for longitudinal behavioral analyses. Each behavioral measure used as the dependent variable (e.g., ADOS, Mullen, and Vineland subscales) is noted at the top of each table. Degrees of freedom, F-statistics, and p-values are reported for main effects of age, subgroup and the age*subgroup interaction. Red stars indicate effects that pass FDR q < 0.05 for multiple comparisons.

• Supplementary file 2. Statistics for each functional connectivity comparison. This file shows statistics for case-control or subtype models for each functional connectivity component-pair comparison. Case-control and subtype models report degrees of freedom, F-stats, p-values, and $\eta^2$, and FDR statistics. Additionally, there are columns reporting the AIC values for the case-control or subtype models. The final set of columns report t-stats, p-values, and effect size (Cohen's d) for specific pairwise group comparisons. Note that these statistics are estimated based on the full dataset of n = 195 toddlers with rsfMRI scanning data available (with or without eye tracking data). Red color indicates comparisons that pass FDR q < 0.05.

• Supplementary file 3. Statistics for pairwise group comparisons from DMN-OTC, DMN-PVC, and DMN-DAN component-pairs. This file shows t-statistics, p-values estimated from non-parametric permutation tests, FDR q-values, and standardized effect sizes (Cohen's d) statistics for each pairwise group comparisons for DMN-OTC, DMN-PVC, and DMN-DAN component pairs. Note that

these statistics are estimated based on the dataset of toddlers that had both rsfMRI and eye tracking data available (n = 164). Red color indicates comparisons that pass FDR q < 0.05.

• Transparent reporting form

### Data availability

Tidy data and analysis code are available at https://github.com/mvlombardo/geoprefrsfmri (copy archived at https://github.com/elifesciences-publications/geoprefrsfmri).

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
