## [Decision Letter]

**Acceptance summary:**

This study reports important eye tracking and resting state fMRI results in a large cohort of toddlers with autism spectrum disorder (ASD). The authors used eye tracking to identify an ASD subtype characterized by early social visual engagement difficulties (GeoPref ASD) and can show that this is related to marked default mode network (DMN)-occipito-temporal cortex (OTC) hypoconnectivity. This DMN-OTC hypoconnectivity is also related to increased severity of social-communication difficulties in GeoPref ASD. This early and pronounced social-visual circuit hypoconnectivity could prove critical for future social-communicative development and could also lead to novel early interventions in these individuals.

**Decision letter after peer review:**

[Editors’ note: the authors were asked to provide a plan for revisions before the editors issued a final decision. What follows is the editors’ letter requesting such plan.]

Thank you for sending your article entitled "Reduced default mode-visual network functional connectivity in autistic toddlers with social engagement difficulties" for peer review at *eLife*. Your article is being evaluated by Christian Büchel as the Senior Editor, a Reviewing Editor, and three reviewers.

Given the list of essential revisions, possibly including new experiments, the editors and reviewers invite you to respond with an action plan and timetable for the completion of the additional work. We plan to share your responses with the reviewers and then issue a binding recommendation.

As you will see from the reviews below the main issues raised by the reviewers are the following:

1) Somewhat arbitrary choice of ADOS scale items.

This seems to be a somewhat arbitrary collection of items from the ADOS which needs to be clarified and motivated.

2) Problematic statistics for the comparison of the GeoPref ASD group (n=16) and the relatively large nonGeo ASD group (n=62).

The authors are encouraged to provide further evidence e.g. by using a non-parametric approach (e.g. bootstrapping).

3) Problematic model comparison using AIC.

The comparison of the stratified subtype model and the traditional unstratified case-control model with regards to DMN-OTC is an important finding. AIC as used can be problematic and the authors are encouraged to provide additional evidence for model superiority e.g. using BIC.

*Reviewer #1:*

The study by Lombardo et al. builds on their previous behavioral research which convincingly identified a subgroup of children with ASD who show a strong preference for dynamic, non-social geometric visual stimuli relative to videos of children in motion. Here, they examine whether there are specific neural signatures for this subgroup of children, referred to as "GeoPref ASD," relative to children with ASD who show a preference for social stimuli as well as several groups of control children. The authors report that the GeoPref ASD group has: (1) reduced social engagement, measured with item scores from the ADOS associated with primarily visual social functions, (2) reduced connectivity between occipital-temporal cortex (OTC) and the default mode network (DMN), and (3) subtype-specific associations for DMN-OTC connectivity.

This study represents an exciting and important direction for this line of research, and the imaging component of this work has many strengths: the extent of resting data collected in toddlers with ASD is impressive, and the brain networks approach is sophisticated and well-conceived. Unfortunately, there are a few critical issues that preclude a more favorable review here. First, there is concern with the social engagement subscale of the ADOS that the authors have used. On its surface, this seems to be a somewhat arbitrary collection of items from the ADOS which apparently has never been used before. For example, it is unclear why other items from the ADOS were not included in the author's social engagement subscale, such as Shared Enjoyment in Interaction and Requesting. Given the centrality of the social engagement measure for all of the brain and behavioral analyses, its arbitrariness and unknown clinical value represent substantial limitations of this work. Furthermore, it is misleading to state that "the GeoPref ASD subtype distinction is indeed a distinction that generalizes beyond eye tracking tests and shows differentiation in early social engagement difficulties, as measured by gold standard symptom based diagnostic measures" (subsection “Behavioral and developmental characteristics of the GeoPref ASD subtype”). While the author's social engagement measure is certainly derived from the gold standard symptom based diagnostic measure (i.e. the ADOS), it currently has no known clinical value. It would have been preferable if the authors had used a more established subscale from the ADOS, such as the social affect subscale, or another measure (e.g. SRS) as the primary behavioral measure here.

A second issue is related to the group comparison performed between the GeoPref ASD and the nonGeo ASD groups for the DMN-OTC connectivity data (i.e., Supplementary file 3), which is a critical one for the hypotheses detailed in this work. As noted by the authors, the sample size of the GeoPref ASD group is small (n=16), and the methods are not clear about what statistical method was used for this comparison. Given the distribution of DMN-OTC connectivity scores in Figure 2, which shows a high degree of overlap in DMN-OTC connectivity between the small GeoPref ASD group and the relatively large nonGeo ASD group (n=62), the *p* = 0.01 group difference result is surprising, and the statistic used to compute this comparison is a critical omission. I imagine this would not survive non-parametric testing in the event the n=16 group is not normally distributed, which the authors do not report.

A third issue is found in the Results where the authors compare the stratified subtype model and the traditional unstratified case-control model with regards to DMN-OTC, which is a critical result for this work. The authors used the Akaike Information Criteria (AIC) statistic to evaluate these two models and, according to the literature, "the individual AIC values are not interpretable as they contain arbitrary constants and are much affected by sample size" (Burnham and Anderson, 2004). Therefore ΔAIC between models needs to be computed. The authors did not report ΔAIC results. Moreover, when they are computed the key result for the DMN-OTC connectivity between stratified subtype model and the traditional unstratified case-control model yields ΔAIC = 3.72. Since Burnham and Anderson state that ΔAIC between 4-7 do not show strong support, it appears that these AIC results are not particularly compelling. It is possible that I am missing something here however it should be noted that the AIC methods are very brief and lack citations.

A fourth issue is related to a strongly worded statement regarding the behavioral data that I am not convinced is warranted given the results. The authors state that "Developmentally, the social engagement and other developmental domains show trajectories indicative of poorer development in GeoPref ASD" (subsection “Behavioral and developmental characteristics of the GeoPref ASD subtype”). This statement about trajectories would suggest that there is an [Age x Subgroup] interaction in the ANOVA results, however in the same paragraph it is stated that there is "no strong evidence of differences in slope of developmental trajectories between the two subtypes" (i.e., no [Age x Subgroup] interaction, Supplementary file 1). If there is no interaction, no claims regarding differential trajectories can be made.

*Reviewer #2:*

In this most interesting paper the authors describe their efforts of detecting an ASD subgroup that displays a specific and extreme lack of preference for socially compelling stimuli as measured by an eye tracking preferential looking paradigm (GeoPref). In previous studies the authors have demonstrated the utility of this task to separate subgroups within the ASD group, but also in comparison to non-ASD control groups with an impressively high specificity.

In the study here described the authors go on to relate the behavioral read out of their task to functional connectivity as assessed by resting state fMRI in toddlers with ASD and other control groups. Importantly, the authors also report efforts which were undertaken to investigate the clinical characteristics of the GeoPref ASD subgroup. Here, it was found that the GeoPref subgroup, indeed, shows more pronounced social impairments as measured by a selected group of ADOS items that the authors claim taps into social engagement difficulties. fMRI results demonstrate the ASD group as a whole exhibits connectivity reduction of the so-called default mode network (DMN) with visual and attention-related brain areas. The authors than included GeoPref for further stratification and report that this improves the findings for DMN-visual cortex connectivity. Finally, they also used a transdiagnostic model which used fixation data as the primary predictor and found that it does not outperform the subtype model.

Taken together, this is an exciting paper that describes advances in the characterization of a subgroup with ASD at the neural level.

1) How were the ADOS items chosen? What is the empirical evidence that makes it plausible to choose these particular items and not others?

2) Did the authors also compare subgroup-specific differences for the communication and social interaction scales of the ADOS? Does the GeoPref subgroup have higher ADOS scores overall?

3) Did the authors also explore the relationship of the GeoPref findings with parental report?

4) Recently, it has been suggested (Schilbach, 2019) that psychiatry needs to move towards observer-independent characterizations of social interaction behavior (e.g. via full body motion tracking) to provide a more comprehensive and clinically relevant analysis. Could this approach have helped the here described study? Could interaction-based phenotyping have helped the transdiagnostic analysis, because fixation data is too limited?

5) What is the clinical relevance of the additional fMRI results? It's great to see that the behavior differences are reflected at the level of the brain and it helps to provide a better characterization of the subgroup, but will this be relevant in the clinic?

6) In the comparison of ASD subtypes surrounding Figure 3 it might be helpful to include the data from TD toddlers to indicated how similar they are to the less affected ASD group.

7) Are the connectivity findings consistent with recent results from Dukart et al. (2019)?

8) The discussion of DMN findings in relation to social cognition and social interaction appears somewhat limited. Many other papers have addressed the relationship of physiological baseline of the brain and the psychological baseline of thinking about others and its relevance for different psychiatric disorders (e.g. Schilbach et al., 2008, Mars et al., 2012, Schilbach et al., 2015, Spreng, 2015).

9) There are recent findings for a subspecialization of different DMN nodes (Kernbach et al., 2018), which raises the question whether other targeted analyses could have been useful to further corroborate the current findings.

10) In my personal opinion the Discussion could be shortened, because it includes a relatively long coverage of putative genetic mechanisms and ideas for future research (iPSC), which are all very interesting, but not closely connected to the findings of the study.

*Reviewer #3:*

Lombardo, Pierce and colleagues provide cognitive experiments with eye tracking that build on a recently discovered eye-tacking based autism "subtype" – an eye tracking preferential looking paradigm. The authors thus replicate and extend this potential subgroup marker in a large independent sample (n=334). This work may contribute exciting new evidence on dysregulation between social visual engagement and the long-standing interest in higher social cognition deficits, like perspective taking, probably subserve by the default mode network at the earliest periods of development. Although this reviewer is favorable of the work, several shortcomings should be addressed for publication in *eLife*.

- Materials and methods section: The implementation of which linear model package was used is explained. However, several aspects of the non-imaging data analysis appear to be missing. Please include more details, including whether or not data columns have been z-scored, which dependent and independent variables are been fed into the model based on what rational.

A similar observation is made by this reviewer about the linear model applied to the partial correlation lower-triangle matrices – it remains largely unclear how the 'stratified' vs. 'unstratified' analysis models were set up. Please specify the exact input variables, output variables, including transformations and potential regularization schemes. Given the high number of component-component connectivity strengths, it is relevant to the reader how linear models were exactly estimated in this scenario of high dimensionality. Finally, how many separate models were estimated in each of these scenarios?

In short, mostly mentioning that the 'lm' function was used is insufficient, given that a large majority of quantitative models fit in the behavioral and biological sciences are some form of linear or generalized linear model.

- Discussion section: Please provide reflection and weighing of the results and findings given the uneven division of subject split into n=16 GeoPref ASD toddlers (11 male, 5 female) and n=62 nonGeo ASD. 16 may appear small, compared to the 195 overall participants.

- Potentially selective citations: other authors have built a body of work on gaze cognition and the brain, such as Schilbach et al., Vogeley et al., and other. The interpretation and introduction may profit from a more balanced relation of the current investigation to existing work.

- Materials and methods section: There may be a slight misunderstanding behind the meaning of bootstrapping as expressed in "to give ranges around the sample correlation estimates": the added value of running a bootstrap analysis in the present context relies on inference on the distribution of outcomes in participant samples that one could have observed.

- Discussion section: The conclusion of "default mode network is functionally disconnected with visual cortices and dorsal attention network on average in ASD" may be overstretched or at least imprecise. Some readers may take this as meaning that DMN and visual cortex are not connected at all in autism, while the authors found a statistically different strengths of functional connectivity between the DMN visual-related components.

[Editors’ note: formal revisions were requested, following approval of the authors’ plan of action.]

Thank you for providing a revision plan for your article titled “Reduced default mode-visual network functional connectivity in autistic toddlers with social engagement difficulties”. We are pleased to inform you that the editors and reviewers have approved your revision plan and look forward to receiving your revised article when ready. After assessing your response, the reviewers had some additional queries, copied below for reference. In your final "response to the reviewers" document, please also address these comments.

Additional essential revisions:

1) I feel that only established measures of social communication should be reported for individuals with ASD: using standard instruments is critical for interpretable clinical research. Therefore, I would greatly prefer if they did not report their novel measure of Social Engagement. Plus, I would like to verify that their "ADOS CoSo Total," an abbreviation that I am not familiar with, is the same as the Social Affect subscore of the ADOS, which is an established measure of social communication for individuals with ASD.

2) Given the small n (n=16) and the fact that Figure 2 appears to show outliers (see whiskers on box plots), a non-parametric 2-sample t-test (Mann-Whitney) seems appropriate.

3) I am confused by the author's response about interpretation of AIC. This is from the Burnham and Anderson (2004) paper they cite:

"The larger the Δ_i_, the less plausible is fitted model i as being the best approximating model in the candidate set....Some simple rules of thumb are often useful in assessing the relative merits of models in the set: Models having Δ_i_ ≤ 2 have substantial support (evidence), those in which 4 ≤ Δ_i_ ≤7 have considerably less support, and models having Δ_i_ > 10 have essentially no support. These rough guidelines have similar counterparts in the Bayesian literature (Raftery 1996)."

My understanding is that this means that the ΔAIC_i_ reported by the authors (3.7) shows "considerably less support" that the fitted model is the best approximating model. I am not an expert in AIC so maybe I am missing something here.

[Editors’ note: this article was then rejected after discussions between the reviewers, but the authors were invited to resubmit after an appeal against the decision.]

Thank you for re-submitting your work entitled "Default mode-visual network hypoconnectivity in an autism subtype with pronounced social visual engagement difficulties" for consideration by *eLife*. Your article has been reviewed by a Senior Editor, a Reviewing Editor, and three reviewers.

Our decision has been reached after consultation between the reviewers. Based on these discussions and the individual reviews below, we regret to inform you that your work will not be considered further for publication in *eLife*.

As you can see from the individual comments all reviewers found the data-set to be very interesting and unique. However, the revisions did not convince two of the three reviewers as you can see from their comments. Robustness of findings is of great importance and especially reviewer #1 continues to be concerned about the statistical procedures, which resonates with the comments made by reviewer #3.

*Reviewer #1:*

While I appreciate that the authors removed the social engagement subscale, there are several critical items from my previous review of this work that I do not feel were sufficiently addressed in this resubmission:

1) It is not clear to me why a non-parametric 2-sample t-test (Mann-Whitney) test, the standard statistical measure in the context of a small n and outliers, was not performed on the critical GeoPref vs. nonGeo ASD group comparisons. This test was specifically requested in my previous review. If this is a robust brain signature for GeoPref ASD, this result should be significant using a Mann-Whitney measure as well as the reported bootstrapping results.

2) I remain concerned about the ΔAIC results. Based on Burnham and Anderson's heuristic, I would expect that strong support for the subtype model would be accompanied by a larger ΔAIC. The reported ΔAIC result (3.7) was not within the range for "considerably less support" (i.e., ΔAIC between 4-7) and is certainly well below the ΔAICs associated with "essentially no support." I would suggest that the editor consults with an expert on AIC to help resolve this issue.

3) I remain concerned about the weak theoretical framing surrounding the DMN-OTC results, a point that I highlighted in my initial review of this work. The DMN is typically considered a task-negative network and understanding its link with task-positive OTC in neurotypical individuals or individuals with ASD is not mentioned by the authors. This is critical given the importance of the DMN-OTC results for this paper.

*Reviewer #2:*

The authors have satisfactorily addressed all, but one of my previous comments:

I still feel that the discussion of putative genetic mechanisms is interesting, but not adequate in light of the results reported.

*Reviewer #3:*

This reviewer thanks the authors for their efforts on this first revision of the manuscript. Several of my core concerns of the previous version of the manuscript revolved around details and inconsistencies in the statistical modeling.

As one example from the revised manuscript:

"Both GLM models were implemented with the lm function in R. In these GLMs the partial correlation for a particular component pair was the dependent variable. […] The primary dependent variable of interest was the group variable."

Here, the authors first mention the connectivity strengths as the dependent variable first, and then instead mention the group variable. This, and other remaining problems (explanation of bootstrapping etc.) make me doubt that the results and conclusion stand on solid ground – which I have already expressed in detail regarding the previous version of the manuscript.

[Editors' note: further revisions were requested prior to acceptance, as described below.]

Thank you for resubmitting your work entitled "Default mode-visual network hypoconnectivity in an autism subtype with pronounced social visual engagement difficulties" for further consideration by *eLife*. Your appeal has been evaluated by Christian Büchel (Senior Editor) and two statistical Editors.

I am sorry that this took so long, but after consulting with one statistical editor, 2 of 3 reviewers remained skeptical and were still against publication of your paper and I therefore had to get a second statistical opinion.

In essence, both statistical editors agree with your line of reasoning concerning sampling procedures and the use of ΔAIC. Therefore, I am very happy to tell you that we are willing to consider publishing your paper. However, during the statistical review of your paper, it has been flagged that the paper often suggests causal relationships indicated by using the verb "drive" (including the abstract). However, the data do not allow this interpretation, because you are simply observing a correlation. This needs to be changed before acceptance.

---

## [Author Response]

[Editors’ note: what follows is the authors’ plan to address the revisions.]

As you will see from the reviews below the main issues raised by the reviewers are the following:1) Somewhat arbitrary choice of ADOS scale items.This seems to be a somewhat arbitrary collection of items from the ADOS which needs to be clarified and motivated.

We can see from the detailed reviewer comments what motivates this first issue here regarding the choice of ADOS scale items that go into the social engagement index we have used. In summary, our response and action plan for this issue is that we can default back to simply using the ADOS social-communication total for the connectivity-behavior relationship analysis. Reviewer 1 had suggested this for the analysis of connectivity-behavioral relationships, and we don’t have any arguments against this. We would still like to use the social engagement index as an independent behavioral validation of differences between the eye tracking defined subtypes, since we feel it is an important point to show that the subtypes are also differentiated on a symptom-based index derived from a gold standard diagnostic measure like the ADOS. In terms of adding items to this index, we can also integrate the two items that reviewer 1 had suggested (shared enjoyment in interaction and requesting). Finally, in a revision, we can immediately add more details about the justifications for selecting these particular items for the social engagement index. In terms of a timetable for implementing these changes, we have already implemented them, and can insert them into a revision immediately.

Below you will find more detailed justifications that specifically answer this first point about what was the rationale for selecting those specific items. The quick answer is that we selected them based on face validity around the construct of social visual engagement. Below you can also find the statistical results for the connectivity-behavior relationships when we used a revised version of the social engagement index (incorporating shared enjoyment in interaction and requesting) and when the ADOS social-communication total was used.

– Justification for selection of specific items for the social engagement index

In the manuscript, we selected items from the ADOS for the social engagement index on the basis of their face validity in being relevant to the construct of social visual engagement. Social visual engagement was the construct under study, given the emphasis on the eye tracking defined subtypes. We had originally called it a “social orienting index”, but changed the label to “social engagement” due to other studies in the literature opting for this label rather than social orienting. Since not all items are purely visual in nature and may require other aspects of behavior (e.g., communication), we were being conservative in referring to this index simply as a “social engagement” index, rather than only being something pertaining purely to visual social engagement. However, our primary intention was for this to mainly be something pertaining to early social visual engagement behaviors.

Items such as eye contact and integration of eye gaze in social overtures are clear items relevant to visually attending to social world. Pointing, initiating and responding to joint attention, and showing are all relevant with regards to the manipulation of attention of self or others and/or the joint sharing of attention/focus between self and other. Response to name was the final key item that was chosen because it signifies attention to the social world by attending to auditory stimuli and then using such stimuli to behave in a way relevant to the stimulus (i.e. turn and look at the person calling their name). Of the total set of items on the ADOS, these were the select items that fulfilled our view of what we think is most relevant to “social visual engagement”, defined as attending and focusing on the social world around the child.

Other items that the reviewers had suggested such as shared enjoyment in interaction and requesting might be relevant to social engagement, but are less relevant to social visual engagement per se. To be conservative in the items we chose, the lack of visual engagement in these items is the primary reason why we did not include them. In particular, shared enjoyment in interaction also adds in an affective component (e.g., pleasure). While this is quite an important item, it is not as relevant to the way we had defined social visual engagement as attending and focusing on the social world around the child. Such a definition does not define what type of affect the children must show towards others. Requesting was also not included as it is more of a communicative item involving getting another person to do something for the child. The biggest distinction we can think of between requesting and what we would consider as “social visual engagement” is the difference between requesting and joint attention. Autistic children can show requesting behavior but without the key aspect of social intent to engage and focus on others or attempting to direct the other’s attention for a non-imperative purpose.

Thus, while these were our initial motivations, we can agree with the editor and reviewers that such a subset of items may not necessarily be independent from other “social” items in the sense of being statistically independent. In this sense, one could say our selection of items was arbitrary from a statistical standpoint, since it was not informed on any statistical basis, but rather was based on the face validity of important behaviors in the ADOS relevant to social visual engagement. Therefore, a much broader look at the social and communicative total from the ADOS could be examined. We did not initially examine this with regards to connectivity-behavior relationships in the first instance because our intention with the social engagement index was to first use it as a real-world symptom-based validation of the eye tracking defined subtypes. Second, we believed that an a priori index specific to the domain of social visual engagement was the best way to examine connectivity-behavior relationships. We have no issue with simply defaulting to reporting relationships between the ADOS social-communication total and connectivity. While there is a loss of specificity in the inferences being made, an analysis of relationship between connectivity and ADOS social-communication total would allow us to see if connectivity has an association with the broader symptom domain of which diagnosis is particularly reliant on.

– Results when a revised social engagement index or ADOS social-communication total is used in connectivity-behavior relationships

Below is a table summarizing analyses looking at connectivity-behavior relationships. The column “Prior version” shows the statistics from the last submission. The column “Revised version” shows the ADOS social engagement index when including the items of shared enjoyment in interaction and requesting. The final column “ADOS CoSo Total” shows statistics when the total ADOS score for all social-communicative algorithm items is used. These results show that DMN-OTC connectivity-social communication relationships for GeoPref ASD are just as strong (if not stronger), and thus the correlation here is not necessarily restricted solely to items with relevance to social visual engagement. Since there is no a priori reason to suspect on a statistical basis that the selected items are independent of other items on the ADOS, and because the ADOS algorithm holds a much stronger clinical foothold in past research, we can agree with the editor and reviewers that probably the most principled thing to do is to proceed using the ADOS social-communication algorithm total in these analyses.

Prior versionRevised versionADOS CoSo TotalGeoPref ASD IC10-IC02 (DMN-OTC)r = -0.61, p = 0.025, CI = [-0.97, -0.23]r = -0.67, p = 0.009, CI = [-0.95, -0.32]r = -0.78, p = 0.001, CI = [-0.99, -0.35]nonGeo ASD IC10-IC02 (DMN-OTC)r = 0.03, p = 0.80, CI = [-0.26, 0.33]r = 0.07, p = 0.59, CI = [-0.22, 0.36]r = 0.06, p = 0.64, CI = [-0.20, 0.33]GeoPref ASD IC10-IC05 (DMN-PVC)r = -0.10, p = 0.74, CI = [-0.95, 0.67]r = -0.05, p = 0.85, CI = [-0.88, 0.72]r = 0.04, p = 0.88, CI = [-0.81, 0.75]nonGeo ASD IC10-IC05 (DMN-PVC)r = 0.13, p = 0.32, CI = [-0.17, 0.39]r = 0.11, p = 0.42, CI = [-0.21, 0.39]r = 0.22, p = 0.12, CI = [-0.10, 0.49]GeoPref ASD IC10-IC09 (DMN-DAN)r = -0.09, p = 0.75, CI = [-0.94, 0.57]r = -0.26, p = 0.41, CI = [-0.95, 0.44]r = -0.33, p = 0.25, CI = [-0.83, 0.58]nonGeo ASD IC10-IC09 (DMN-DAN)r = 0.25, p = 0.07, CI = [-0.02, 0.46]r = 0.29, p = 0.035, CI = [0.09, 0.49]r = 0.29, p = 0.045, CI = [0.05, 0.49]

2) Problematic statistics for the comparison of the GeoPref ASD group (n=16) and the relatively large nonGeo ASD group (n=62). The authors are encouraged to provide further evidence e.g. by using a non-parametric approach (e.g. bootstrapping)

We have now used a non-parametric approach to hypothesis testing (e.g., permutation test). After 10,000 permutations, we can assign a p-value to each hypothesis test by comparing the actual t-statistic under the true group labels with the null distribution of t-statistics recomputed when the group labels are randomly shuffled. This approach results in the same inferences as the prior approach with parametric hypothesis tests. See the tables presented below.

Because the results are effectively the same, we leave it to the decision of the editor and reviewers whether you would like us to use the parametric or non-parametric results. In terms of a timetable, these revisions to the analyses have already been coded up and computed and can be included in a revision immediately.

IC10-IC02 (DMN-OTC)

Comparisont-stat (parametric)p-value (parametric)p-value (non-parametric)FDR q (parametric)FDR q (non-parametric)GeoPref ASD vs.nonGeo ASD *-2.65440.01310.01560.02620.0312GeoPref ASD vs.LD/DD *-2.75160.01090.01060.02620.0265GeoPref ASD vs.TypSibASD *-3.63080.00100.00170.00520.0085GeoPref ASD vs.TD *-4.66310.00010.00040.00090.0040nonGeo ASD vs.LD/DD-1.13290.27140.26650.38770.3807nonGeo ASD vs.TypSibASD-1.72130.09640.09390.16070.1565nonGeo ASD vs.TD *-2.70460.00790.00820.02620.0265LD/DD vs.TypSibASD-0.10250.91920.91210.91920.9121LD/DD vs.TD-0.27180.78880.78770.91920.9121TypSibASD vs.TD-0.21820.82900.82380.91920.9121

IC10-IC05 (DMN-PVC)

Comparisont-stat (parametric)p-value (parametric)p-value (non-parametric)FDR q (parametric)FDR q (non-parametric)GeoPref ASD vs.nonGeo ASD-0.54700.58940.59600.65490.6623GeoPref ASD vs.LD/DD *-2.67990.01380.01180.04610.0345GeoPref ASD vs.TypSibASD *-2.28020.03000.02840.05610.0487GeoPref ASD vs.TD *-2.64390.01350.01210.04610.0345nonGeo ASD vs.LD/DD *-2.60460.01890.01380.04730.0345nonGeo ASD vs.TypSibASD *-2.26360.03360.02920.05610.0487nonGeo ASD vs.TD *-3.06790.00270.00250.02710.0250LD/DD vs.TypSibASD1.04240.30810.31020.38510.3877LD/DD vs.TD1.31880.20470.20020.29240.2860TypSibASD vs.TD0.25470.80110.80590.80110.8059

IC10-IC09 (DMN-DAN)

Comparisont-stat (parametric)p-value (parametric)p-value (non-parametric)FDR q (parametric)FDR q (non-parametric)GeoPref ASD vs.nonGeo ASD-1.03270.31280.31480.39100.3935GeoPref ASD vs.LD/DD-2.32830.02710.02820.09040.0940GeoPref ASD vs.TypSibASD-1.76280.08870.08680.17750.1736GeoPref ASD vs.TD *-2.85170.00850.00900.04240.0450nonGeo ASD vs.LD/DD-1.93400.06710.06310.16770.1577nonGeo ASD vs.TypSibASD-1.19090.24380.24040.39100.3935nonGeo ASD vs.TD *-2.86840.00500.00550.04240.0450LD/DD vs.TypSibASD0.86340.39570.40160.43960.4462LD/DD vs.TD0.05350.95780.95680.95780.9568TypSibASD vs.TD-1.09190.28280.28280.39100.3935

3) Problematic model comparison using AIC.The comparison of the stratified subtype model and the traditional unstratified case-control model with regards to DMN-OTC is an important finding. AIC as used can be problematic and the authors are encouraged to provide additional evidence for model superiority e.g. using BIC.

To summarize our response on this issue, we argue that AIC is the best measure of model selection criteria in this specific context, particularly with respect to BIC. We also would like to thank the reviewer for pointing out that we should have gone beyond simply evaluating AIC based on which model produced the lowest AIC. Although the winner under that rule was the subtype model, ΔAIC is further evidence we should report. However, contrary to the ΔAIC measure showing that “the results are not that compelling”, this ΔAIC measure actually shows the opposite – there is indeed evidence to support that the case-control model is not as compelling a model as the best performing subtype model. As such, we do not see the need to use other metrics of model selection criteria (e.g., BIC). As action plan, we think it is appropriate to add to the manuscript the reviewer’s suggestion of the ΔAIC measure. This action can be implemented in a revision immediately. Below is a much more detailed response on each of these points.

– Using AIC, not BIC, in the case of complex “tapering effects”

We have used the AIC as a criterion for selecting whether the best model is a case-control or subtype model. We have not utilized other criteria like BIC because as stated by Burnham and Anderson (2004), BIC is best when one suspects that the true model of reality is one characterized by a few big effects, rather than complex effects (what Burnham and Anderson (2004) call “tapering effects”). Indeed, one philosophical problem about BIC’s use in practice is this assumption that the true model of reality be one that is not complex, since it is hardly ever the case that reality is not complex, at least with respect to phenotypes like autism. Burnham and Anderson (2004) exemplify this difference in assumptions about the nature of the true model by contrasting Figure 1 vs. 2 in their paper. BIC works best under the situation of their Figure 2 where the ground truth is that there are only a few large effects, and all other effects are negligible. AIC works best when the ground truth is likely indicative of complexity (or “tapering effects”) as shown in their Figure 1. Illustrating this with the exact words of Burnham and Anderson (2004):

“It is known (from numerous papers and simulations in the literature) that in the tapering-effects context (Figure 1), AIC performs better than BIC. If this is the context of one’s real data analysis, then AIC should be used” (Burnham and Anderson, 2004, pg. 285).

Given our current understanding of autism, the reality is likely one of complexity. In fact, the very reason for studying heterogeneity is because the effects underlying the label of autism are likely complex. There is no reason to suspect the underlying truth (i.e. full model) for explaining features of autism will be a simplistic model with few very large effects (see Happe, Ronald, & Plomin, 2006). For this reason, AIC is preferred as the best measure of model selection criterion in this situation, especially when compared to BIC.

The other issue to illustrate for why BIC should not be utilized is because the computation behind BIC can be considered as mathematically identical to AIC, but with a primary difference in the penalty parameter (*k*) for model complexity. For AIC, *k* = 2, whereas for BIC *k* = log(n) and is typically much larger than *k* for AIC. Burnham and Anderson (2004) describe this difference as basically an argument that reduces down to the priors on the models (and they illustrate in their paper that AIC is just as Bayesian as BIC). BIC has a prior that the best model should be more simplistic with few large effects, hence a larger penalty parameter for complexity, whereas AIC has the prior that the model should likely be complex, and hence there is a smaller penalty parameter for complexity of the model. It makes little logical sense for us to use AIC and then compute BIC again, which is simply another computation of AIC with a different penalty parameter that implicitly assumes the true model should be simple with few large effects. For all the reasons mentioned above, this is why we prefer not to confuse the situation by reporting both model selection criterion. Had we needed to provide the BIC for the DMN-OTC models, it would have shown that the subtype model produces a lower BIC than the case-control model for DMN-OTC (subtype BIC = -26.53; case-control BIC = -25.90). As Burnham and Anderson (2004) note, the best model to select for BIC is the one producing the lowest BIC value. Hence, even here with BIC, the overall result is the same as AIC in showing that the subtype model is better than the case-control model. However, we do not see how this is necessarily helpful and actually only introduces more confusion, as conceptually there are issues with using BIC in our context when we are fairly certain the situation in autism is contrary to the expectations of BIC.

– ΔAIC supports the subtype model is better than the case-control model

Finally, we also would like to point out that contrary to the reviewer comment about ΔAIC, this evidence actually supports the idea that the subtype model is indeed a better model and that there is considerably less support for the idea that the case-control model is as good. Burnham and Anderson (2004) outline that ΔAIC is computed as follows: ΔAIC_i_ = AIC_i_ – AIC_min_. AIC_min_ is the AIC for the model with the lowest AIC – that is, the subtype model for DMN-OTC. AIC_i_ is then the AIC for the case-control model. ΔAIC_i_ is the metric used to make an inference about whether there is evidence supporting that the case-control model (AIC_i_) is as good as the subtype model (AIC_min_). The ΔAIC_i_ = 3.72. Burnham and Anderson (2004) give this heuristic advice for interpreting ΔAIC_i_:

“Models having Δ_i_ ≤ 2 have substantial support (evidence), those in which 4 ≤ Δ_i_ ≤ 7 have considerably less support, and models having Δ_i_ > 10 have essentially no support”(Burnham and Anderson, 2004, pg. 271).

In other words, a model compared to the model with the lowest AIC (i.e. the best model) that results in 4 ≤ Δ_i_ ≤ 7 has considerably less support than the best model with the lowest AIC. This means there is “considerably less support” for the case-control model (AIC_i_) compared to the subtype model (AIC_min_). This contradicts the statement in review 1, *“*it appears that these AIC results are not particularly compelling***”***. The opposite is actually the case – the evidence here suggests that there is considerably less support for the case-control model being as good as the best model (AIC_min_), the subtype model. Therefore, AIC is the best model selection criterion for us to use in this context, and the evidence is clear that the subtype model is a better model than the case-control model for the DMN-OTC comparison.

[Editors’ notes: the authors’ response after being formally invited to submit a revised submission follows.]

As you will see from the reviews below the main issues raised by the reviewers are the following:1) Somewhat arbitrary choice of ADOS scale items.This seems to be a somewhat arbitrary collection of items from the ADOS which needs to be clarified and motivated.

Per the request of the reviewers, in the revision we have now used the ADOS social affect total and have dropped the usage of an ADOS social engagement index.

2) Problematic statistics for the comparison of the GeoPref ASD group (n=16) and the relatively large nonGeo ASD group (n=62). The authors are encouraged to provide further evidence e.g. by using a non-parametric approach (e.g. bootstrapping)

We have now used a non-parametric approach to hypothesis testing (e.g., permutation test). After 10,000 permutations, we can assign a p-value to each hypothesis test by comparing the actual t-statistic under the true group labels with the null distribution of t-statistics recomputed when the group labels are randomly shuffled. This approach results in the same inferences as the prior approach with parametric hypothesis tests. See the tables presented in our revision plan. As suggested by the reviewers we utilize these non-parametric results in the revised manuscript.

3) Problematic model comparison using AIC.The comparison of the stratified subtype model and the traditional unstratified case-control model with regards to DMN-OTC is an important finding. AIC as used can be problematic and the authors are encouraged to provide additional evidence for model superiority e.g. using BIC.

Regarding whether to use AIC or BIC, Burnham and Anderson (2004) noted:

“It is known (from numerous papers and simulations in the literature) that in the tapering-effects context (Figure 1), AIC performs better than BIC. If this is the context of one’s real data analysis, then AIC should be used”.

Burnham and Anderson (2004) use the terminology of “tapering effects” to mean a context with high complexity. In these situations, AIC is preferred over BIC. BIC is preferred in situations where the true reality is one where a few large effects dominate. Given our current understanding of autism, the reality is likely one of complexity. In fact, the very reason for studying heterogeneity is because the effects underlying the label of autism are likely complex. There is no reason to suspect the underlying truth (i.e. full model) for explaining features of autism will be a simplistic model with few very large effects (see Happe, Ronald and Plomin, 2006). For this reason, AIC is preferred as the best measure of model selection criterion in this situation, especially when compared to BIC.

The other issue to illustrate for why BIC should not be utilized is because the computation behind BIC is mathematically identical to AIC, but with a primary difference in the penalty parameter (*k*) for model complexity. For AIC, *k* = 2, whereas for BIC *k* = log(n) and is typically much larger than *k* for AIC. Burnham and Anderson (2004) describe this difference as basically an argument that reduces down to the priors on the models (and they illustrate in their paper that AIC is just as Bayesian as BIC). BIC has a prior that the best model should be more simplistic with a few large effects, hence a larger penalty parameter for complexity, whereas AIC has the prior that the model should likely be complex, and hence there is a smaller penalty parameter for complexity of the model. It makes little logical sense for us to use AIC and then compute BIC again, which is simply another computation of AIC with a different penalty parameter that implicitly assumes the true model should be simple with a few large effects.

Finally, we also would like to point out that contrary to the reviewer comment about ΔAIC, there is not compelling evidence that the case-control model is as good as the subtype model. Burnham and Anderson (2004) outline that ΔAIC is computed as follows: ΔAIC_i_ = AIC_i_ – AIC_min_. AIC_min_ is the AIC for the model with the lowest AIC – that is, the subtype model for DMN-OTC. AIC_i_ is then the AIC for the case-control model. ΔAIC_i_ is the metric used to make an inference about whether there is evidence supporting that the case-control model (AIC_i_) is as good as the subtype model (AIC_min_). The ΔAIC_i_ = 3.72. Burnham and Anderson (2004) give this heuristic advice for interpreting ΔAIC_i_:

“Models having Δ_i_ ≤ 2 have substantial support (evidence), those in which 4 ≤ Δ_i_ ≤ 7 have considerably less support, and models having Δ_i_ > 10 have essentially no support.”

In other words, a model compared to the model with the lowest AIC (i.e. the best model) that results in 4 ≤ Δ_i_ ≤ 7 has considerably less support than the best model with the lowest AIC. The observed ΔAIC_i_ is closest to the above situation, and this would mean that there is “considerably less support” for the case-control model (AIC_i_) compared to the subtype model (AIC_min_). This contradicts the statement in review 1, “it appears that these AIC results are not particularly compelling”. The opposite is actually the case – the evidence here suggests that there is considerably less support for the case-control model being as good as the best model (AIC_min_), the subtype model.

1) I feel that only established measures of social communication should be reported for individuals with ASD: using standard instruments is critical for interpretable clinical research. Therefore, I would greatly prefer if they did not report their novel measure of Social Engagement. Plus, I would like to verify that their "ADOS CoSo Total," an abbreviation that I am not familiar with, is the same as the Social Affect subscore of the ADOS, which is an established measure of social communication for individuals with ASD.

We now utilize the ADOS Social Affect score and have dropped the social engagement index. Our abbreviation of ADOS CoSo Total is indeed the same as the Social Affect subscore.

2) Given the small n (n=16) and the fact that Figure 2 appears to show outliers (see whiskers on box plots), a non-parametric 2-sample t-test (Mann-Whitney) seems appropriate.

As noted above, we have now utilized the non-parametric permutation tests as the primary tests of the specific between-group comparisons.

3) I am confused by the author's response about interpretation of AIC. This is from the Burnham and Anderson (2004) paper they cite:"The larger the Δ_i_, the less plausible is fitted model i as being the best approximating model in the candidate set....Some simple rules of thumb are often useful in assessing the relative merits of models in the set: Models having Δ_i_ ≤ 2 have substantial support (evidence), those in which 4 ≤ Δ_i_ ≤ 7 have considerably less support, and models having Δ_i_ > 10 have essentially no support. These rough guidelines have similar counterparts in the Bayesian literature (Raftery 1996)."My understanding is that this means that the ΔAIC_i_ reported by the authors (3.7) shows "considerably less support" that the fitted model is the best approximating model. I am not an expert in AIC so maybe I am missing something here.

ΔAIC_i_ is a metric indicating whether model i shows compelling evidence for being a model as good as the model with the lowest AIC (e.g., AIC_min_). In this study, AIC_min_ is the subtype model. Thus, ΔAIC_i_ indicates whether the case-control model is as good of a model as the subtype. The heuristic suggested by Burnham and Anderson is that 4 ≤ ΔAIC_i_ ≤ 7 indicates that there is considerably less support for model i (e.g., the case-control model) being as good as the best model (e.g., subtype model). Our ΔAIC_i_ = 3.72, and this is closest to the aforementioned heuristic suggested by Burnham and Anderson (2004). This ΔAIC_i_ would not fit for other heuristics (i.e. ΔAIC_i_ ≤ 2) whereby model i (the case-control model) has substantial support for being as good as the best model.

Reviewer #1:[…] Unfortunately, there are a few critical issues that preclude a more favorable review here. First, there is concern with the social engagement subscale of the ADOS that the authors have used. On its surface, this seems to be a somewhat arbitrary collection of items from the ADOS which apparently has never been used before. For example, it is unclear why other items from the ADOS were not included in the author's social engagement subscale, such as Shared Enjoyment in Interaction and Requesting. Given the centrality of the social engagement measure for all of the brain and behavioral analyses, its arbitrariness and unknown clinical value represent substantial limitations of this work. Furthermore, it is misleading to state that "the GeoPref ASD subtype distinction is indeed a distinction that generalizes beyond eye tracking tests and shows differentiation in early social engagement difficulties, as measured by gold standard symptom based diagnostic measures" (subsection “Behavioral and developmental characteristics of the GeoPref ASD subtype”). While the author's social engagement measure is certainly derived from the gold standard symptom based diagnostic measure (i.e., the ADOS), it currently has no known clinical value. It would have been preferable if the authors had used a more established subscale from the ADOS, such as the social affect subscale, or another measure (e.g., SRS) as the primary behavioral measure here.

Per the request of the reviewers, in the revision we have now defaulted back to simply using the ADOS social affect total instead of the social engagement index.

A second issue is related to the group comparison performed between the GeoPref ASD and the nonGeo ASD groups for the DMN-OTC connectivity data (i.e., Supplementary file 3), which is a critical one for the hypotheses detailed in this work. As noted by the authors, the sample size of the GeoPref ASD group is small (n=16), and the methods are not clear about what statistical method was used for this comparison. Given the distribution of DMN-OTC connectivity scores in Figure 2, which shows a high degree of overlap in DMN-OTC connectivity between the small GeoPref ASD group and the relatively large nonGeo ASD group (n=62), the p = 0.01 group difference result is surprising, and the statistic used to compute this comparison is a critical omission. I imagine this would not survive non-parametric testing in the event the n=16 group is not normally distributed, which the authors do not report.

We have now used a non-parametric approach to hypothesis testing (e.g., permutation test). After 10,000 permutations, we can assign a p-value to each hypothesis test by comparing the actual t-statistic under the true group labels with the null distribution of t-statistics recomputed when the group labels are randomly shuffled. This approach results in the same inferences as the prior approach with parametric hypothesis tests. See the tables presented in the revision plan. As suggested by the reviewers we utilize these non-parametric results for the revised manuscript.

A third issue is found in the Results where the authors compare the stratified subtype model and the traditional unstratified case-control model with regards to DMN-OTC, which is a critical result for this work. The authors used the Akaike Information Criteria (AIC) statistic to evaluate these two models and, according to the literature, "the individual AIC values are not interpretable as they contain arbitrary constants and are much affected by sample size" (Burnham and Anderson, 2004). Therefore ΔAIC between models needs to be computed. The authors did not report ΔAIC results. Moreover, when they are computed the key result for the DMN-OTC connectivity between stratified subtype model and the traditional unstratified case-control model yields ΔAIC = 3.72. Since Burnham and Anderson state that ΔAIC between 4-7 do not show strong support, it appears that these AIC results are not particularly compelling. It is possible that I am missing something here however it should be noted that the AIC methods are very brief and lack citations.

We have now reported ΔAIC in the revised manuscript. However, this result is contrary to the reviewer suggests in this comment. This evidence supports the idea there is not compelling evidence that the case-control model is as good as the subtype model. Burnham and Anderson (2004) outline that ΔAIC is computed as follows: ΔAIC_i_ = AIC_i_ – AIC_min_. AIC_min_ is the AIC for the model with the lowest AIC – that is, the subtype model for DMN-OTC. AIC_i_ is then the AIC for the case-control model. ΔAIC_i_ is the metric used to make an inference about whether there is evidence supporting that the case-control model (AIC_i_) is as good as the subtype model (AIC_min_). The ΔAIC_i_ = 3.72. Burnham and Anderson (2004) give this heuristic advice for interpreting ΔAIC_i_:

“Models having Δ_i_ ≤ 2 have substantial support (evidence), those in which 4 ≤ Δ_i_ ≤ 7 have considerably less support, and models having Δ_i_ > 10 have essentially no support.”

In other words, a model compared to the model with the lowest AIC (i.e. the best model) that results in 4 ≤ Δ_i_ ≤ 7 has considerably less support than the best model with the lowest AIC. The observed ΔAIC_i_ is closest to the above situation, and this would mean that there is “considerably less support” for the case-control model (AIC_i_) compared to the subtype model (AIC_min_). This contradicts the statement in review 1, “it appears that these AIC results are not particularly compelling”. The opposite is actually the case – the evidence here suggests that there is considerably less support for the case-control model being as good as the best model (AIC_min_), the subtype model.

A fourth issue is related to a strongly worded statement regarding the behavioral data that I am not convinced is warranted given the results. The authors state that "Developmentally, the social engagement and other developmental domains show trajectories indicative of poorer development in GeoPref ASD" (subsection “Behavioral and developmental characteristics of the GeoPref ASD subtype”). This statement about trajectories would suggest that there is an [Age x Subgroup] interaction in the ANOVA results, however in the same paragraph it is stated that there is "no strong evidence of differences in slope of developmental trajectories between the two subtypes" (i.e., no [Age x Subgroup] interaction, Supplementary file 1). If there is no interaction, no claims regarding differential trajectories can be made.

We apologize for the confusion in how we have phrased this statement. We intended to state that the GeoPref ASD group is poorer across social engagement and other developmental domains across the age range studied (and as evidenced by the general group effect). We have reworded this part of the Results section in the revision to avoid this confusion.

Reviewer #2:[…]1) How were the ADOS items chosen? What is the empirical evidence that makes it plausible to choose these particular items and not others?

Per the request of the reviewers, in the revision we have now defaulted back to simply using the ADOS social affect total instead of the social engagement index.

2) Did the authors also compare subgroup-specific differences for the communication and social interaction scales of the ADOS? Does the GeoPref subgroup have higher ADOS scores overall?

These analyses were already reported in Figure 1C and Supplementary file 1 and Figure 1—figure supplement 1. The GeoPref ASD subtype did show overall higher levels of severity on the ADOS social affect and the repetitive restricted behaviors total, and there was no difference in the slopes of the trajectories between groups.

3) Did the authors also explore the relationship of the GeoPref findings with parental report?

We did not explore this relationship as parental report measures (e.g., the ADI-R) were not available.

4) Recently, it has been suggested (Schilbach, 2019) that psychiatry needs to move towards observer-independent characterizations of social interaction behavior (e.g. via full body motion tracking) to provide a more comprehensive and clinically relevant analysis. Could this approach have helped the here described study? Could interaction-based phenotyping have helped the transdiagnostic analysis, because fixation data is too limited?

Interaction-based methods would be great to study in the future. We have inserted comments about this in the revision.

5) What is the clinical relevance of the additional fMRI results? It's great to see that the behavior differences are reflected at the level of the brain and it helps to provide a better characterization of the subgroup, but will this be relevant in the clinic?

The fMRI data helps us to better answer hypotheses about whether phenotypically differentiated ASD subtypes are also different on-average at a neural level. We do not have evidence from classifiers or individualized prediction models that utilize rsfMRI data that could answer any questions relevant to real-world applications in clinical settings.

6) In the comparison of ASD subtypes surrounding Figure 3 it might be helpful to include the data from TD toddlers to indicated how similar they are to the less affected ASD group.

ADOS data in TD toddlers shows floor effects, since the ADOS is not sensitive for picking up typically-developing variability and is more tailored to assessing atypical social-communication and RRB behaviors in suspected ASD cases. Thus, because of the floor effects, the data cannot be analyzed in a similar way as those shown with ASD children in Figure 3.

7) Are the connectivity findings consistent with recent results from Dukart et al. (2019)?

Unfortunately, it is not possible to easily compare the current results with those of the recent paper by Holiga et al., 2019. First, the age ranges are completely different, with our dataset examining much younger toddlers with ASD, while Holiga et al. is restricted to examining individuals age 6 years or older. Second, Holiga et al. uses a different metric connectivity – weighted degree centrality (WDC). WDC is incompatible with the ICA between-network method we have used in this paper. With WDC as utilized by Holiga et al., one computes the sum of correlation coefficients between a seed voxel and all other voxels above some arbitrary pre-specified threshold (e.g., r > 0.25). This metric is much more global and less specific than connectivity between specific components or networks, as we have measured in the current study.

8) The discussion of DMN findings in relation to social cognition and social interaction appears somewhat limited. Many other papers have addressed the relationship of physiological baseline of the brain and the psychological baseline of thinking about others and its relevance for different psychiatric disorders (e.g. Schilbach et al., 2008, Mars et al., 2012, Schilbach et al., 2015, Spreng, 2015).

Yes, the DMN is an important network for social cognition, and much of our own past work has demonstrated this and how it may be affected in autism (e.g., Lombardo et al., 2010). We did not expand more on this in the Discussion section for reasons of length of the entire Discussion section. However, we have added more references specific to the topic of DMN regions and their role in autism.

9) There are recent findings for a subspecialization of different DMN nodes (Kernbach et al., 2018), which raises the question whether other targeted analyses could have been useful to further corroborate the current findings.

Certainly, there are ways of splitting the DMN into further subsystems. However, since we were using a data-driven method like ICA, we used the ICA-identified components for further downstream analysis. In the future it may be possible to have methods like ICA or others to search for more granular decompositions of networks and to split the DMN into subsystems. This is beyond the scope of the current investigation, but could be an important topic for future work, and we have specifically noted this in the revision.

10) In my personal opinion the Discussion could be shortened, because it includes a relatively long coverage of putative genetic mechanisms and ideas for future research (iPSC), which are all very interesting, but not closely connected to the findings of the study.

The paragraph discussing more molecular biological mechanisms is something we would like to retain in the Discussion, as it attempts to discuss what may be underlying these neural circuitry differences in ASD subtypes, and possible ways to move forward in future research to get a better grasp on what underlies these differences in living patients.

Reviewer #3:[…]- Materials and methods section: The implementation of which linear model package was used is explained. However, several aspects of the non-imaging data analysis appear to be missing. Please include more details, including whether or not data columns have been z-scored, which dependent and independent variables are been fed into the model based on what rational.

We have now included z-scoring of the data columns in the revision.

In these longitudinal analyses the only independent variables are age and group and the dependent variable for any particular model is whatever developmental measure we are investigating (e.g., Mullen subscales, Vineland subscales, ADOS total scores). We have made this much more explicit in the revision.

For those that are interested, the code for this and all other analyses are laid out in our GitHub repo for this paper: https://github.com/mvlombardo/geoprefrsfmri.

A similar observation is made by this reviewer about the linear model applied to the partial correlation lower-triangle matrices – it remains largely unclear how the 'stratified' vs. 'unstratified' analysis models were set up. Please specify the exact input variables, output variables, including transformations and potential regularization schemes. Given the high number of component-component connectivity strengths, it is relevant to the reader how linear models were exactly estimated in this scenario of high dimensionality. Finally, how many separate models were estimated in each of these scenarios?

The partial correlations matrices were extracted per each individual subject using the MATLAB script nets_netmats.m from the FSLNets MATLAB toolbox (https://fsl.fmrib.ox.ac.uk/fsl/fslwiki/FSLNets) with it set to run ridge regression (Tikhonov-regularization) with the rho parameter set to 1, as we have done in other independent studies (e.g., Lombardo et al., 2019). The input to this function is simply a [time, component] matrix, and the output is a [component, component] partial correlation matrix, where in each cell of that matrix is the partial correlation between that seed component and a target component, partialing out the effects of all other components, using ridge regression for the regularization.

Only one model was implemented. We did not implement separate models. This extraction of partial correlations has nothing to do with the stratification analyses that were done at the group-level, as all of these extractions of partial correlations are done within each subject and all of this occurs before any group-level modeling, which we do in R with the lm function for running a general linear model.

In the revision, we have stated explicitly the input and output variables for the general linear models examining case-control or subtype models. Lombardo et al. (2019).

In short, mostly mentioning that the 'lm' function was used is insufficient, given that a large majority of quantitative models fit in the behavioral and biological sciences are some form of linear or generalized linear model.

The group-level analysis in R was implemented using the lm function in R. The type of linear model lm can run depends on the formula it is given. Here it is a general linear model (GLM), and we have now provided more information in the revised manuscript so that this is clear. The primary independent variable of interest is a factor variable indicating case or control status or a factor variable indicating the specific group labels (e.g., ASD subtype labels), and there are additional covariates of no interest like scan age and sex. After estimating the general linear model, the model object can be passed to the anova function in R in order to extract F-stats and p-values.

- Discussion section: Please provide reflection and weighing of the results and findings given the uneven division of subject split into n=16 GeoPref ASD toddlers (11 male, 5 female) and n=62 nonGeo ASD. 16 may appear small, compared to the 195 overall participants.

In the revision we have discussed more about the uneven divisions between GeoPref and nonGeo ASD.

- Potentially selective citations: other authors have built a body of work on gaze cognition and the brain, such as Schilbach et al., Vogeley et al., and other. The interpretation and introduction may profit from a more balanced relation of the current investigation to existing work.

We have cited this work in the revision.

- Materials and methods section: There may be a slight misunderstanding behind the meaning of bootstrapping as expressed in "to give ranges around the sample correlation estimates": the added value of running a bootstrap analysis in the present context relies on inference on the distribution of outcomes in participant samples that one could have observed.

We have reworded this in the revision.

- Discussion section: The conclusion of "default mode network is functionally disconnected with visual cortices and dorsal attention network on average in ASD." may be overstretched or at least imprecise. Some readers may take this as meaning that DMN and visual cortex are not connected at all in autism, while the authors found a statistically different strengths of functional connectivity between the DMN visual-related components.

We have removed these types of statements in the revision.

[Editors’ note: the author responses to the second round of peer review follow.]

As you can see from the individual comments all reviewers found the data-set to be very interesting and unique. However, the revisions did not convince two of the three reviewers as you can see from their comments. Robustness of findings is of great importance and especially reviewer #1 continues to be concerned about the statistical procedures, which resonates with the comments made by reviewer #3.Reviewer #1:While I appreciate that the authors removed the social engagement subscale, there are several critical items from my previous review of this work that I do not feel were sufficiently addressed in this resubmission:1) It is not clear to me why a non-parametric 2-sample t-test (Mann-Whitney) test, the standard statistical measure in the context of a small n and outliers, was not performed on the critical GeoPref vs. nonGeo ASD group comparisons. This test was specifically requested in my previous review. If this is a robust brain signature for GeoPref ASD, this result should be significant using a Mann-Whitney measure as well as the reported bootstrapping results.

We apologize for the confusion, but it was not clear to us that reviewer 1 was asking for the Mann-Whitney U test as the one and only test of non-parametric inference that would be valid.

The Mann-Whitney U test is a relatively antiquated method for running non-parametric hypothesis testing. It was developed during a time before the advent of fast computers to implement more exact computationally intensive resampling techniques. Most modern statisticians would recommend resampling methods like the permutation test rather than these older tests. This is what we have done.

Please see the full tables below that show the results of Mann-Whitney U tests alongside the original parametric tests and the permutation tests from the last revision. You can see that the Mann-Whitney U results show the same effects as the original parametric hypothesis tests we had initially reported, as well as the non-parametric permutation test results we included in the last revision.

Below are tables that show the results of Mann-Whitney U tests alongside the original parametric tests and the permutation tests from the last revision. In our revised manuscript we do not further include Mann-Whitney U tests, since they do not provide any additional useful information over and above the parametic and non-parametric permutation tests that were included in the previous revision.

IC10-IC02 (DMN-OTC)

Comparisont-stat (parametric)p-value (parameteric)p-value (perm)p-value (Mann-Whitney)FDR q (parametric)FDR q (perm)FDR q (Mann-Whitney)GeoPref ASD vs.nonGeo ASD *-2.65440.01310.01560.01130.02620.03120.0304GeoPref ASD vs.LD/DD *-2.75160.01090.01060.01520.02620.02650.0304GeoPref ASD vs.TypSibASD *-3.63080.00100.00170.00100.00520.00850.0052GeoPref ASD vs.TD *-4.66310.00010.00040.000090.00090.00400.0009nonGeo ASD vs.LD/DD-1.13290.27140.26650.27140.38770.38070.3878nonGeo ASD vs.TypSibASD-1.72130.09640.09390.14250.16070.15650.2375nonGeo ASD vs.TD *-2.70460.00790.00820.01360.02620.02650.0304LD/DD vs.TypSibASD-0.10250.91920.91210.95340.91920.91210.9534LD/DD vs.TD-0.27180.78880.78770.85230.91920.91210.9470TypSibASD vs.TD-0.21820.82900.82380.71530.91920.91210.8941

IC10-IC05 (DMN-PVC)

Comparisont-stat (parametric)p-value (parameteric)p-value (perm)p-value (Mann-Whitney)FDR q (parametric)FDR q (perm)FDR q (Mann-Whitney)GeoPref ASD vs.nonGeo ASD-0.54700.58940.59600.94570.65490.66230.9835GeoPref ASD vs.LD/DD *-2.67990.01380.01180.00820.04610.03450.0273GeoPref ASD vs.TypSibASD-2.28020.03000.02840.03510.05610.04870.0599GeoPref ASD vs.TD *-2.64390.01350.01210.01450.04610.03450.0364nonGeo ASD vs.LD/DD *-2.60460.01890.01380.00190.04730.03450.0106nonGeo ASD vs.TypSibASD-2.26360.03360.02920.03590.05610.04870.0599nonGeo ASD vs.TD *-3.06790.00270.00250.00210.02710.02500.0106LD/DD vs.TypSibASD1.04240.30810.31020.16270.38510.38770.2034LD/DD vs.TD1.31880.20470.20020.04980.29240.28600.0712TypSibASD vs.TD0.25470.80110.80590.98350.80110.80590.9835

IC10-IC09 (DMN-DAN)

Comparisont-stat (parametric)p-value (parameteric)p-value (perm)p-value (Mann-Whitney)FDR q (parametric)FDR q (perm)FDR q (Mann-Whitney)GeoPref ASD vs.nonGeo ASD-1.03270.31280.31480.19170.39100.39350.3195GeoPref ASD vs.LD/DD-2.32830.02710.02820.01700.09040.09400.0569GeoPref ASD vs.TypSibASD-1.76280.08870.08680.06140.17750.17360.1536GeoPref ASD vs.TD *-2.85170.00850.00900.00340.04240.04500.0345nonGeo ASD vs.LD/DD-1.93400.06710.06310.11210.16770.15770.2243nonGeo ASD vs.TypSibASD-1.19090.24380.24040.33750.39100.39350.4219nonGeo ASD vs.TD *-2.86840.00500.00550.00910.04240.04500.0455LD/DD vs.TypSibASD0.86340.39570.40160.62590.43960.44620.6955LD/DD vs.TD0.05350.95780.95680.97710.95780.95680.9771TypSibASD vs.TD-1.09190.28280.28280.32510.39100.39350.4219

2) I remain concerned about the ΔAIC results: Based on Burnham and Anderson's heuristic, I would expect that strong support for the subtype model would be accompanied by a larger ΔAIC. The reported ΔAIC result (3.7) was not within the range for "considerably less support" (i.e., ΔAIC between 4-7) and is certainly well below the ΔAICs associated with "essentially no support." I would suggest that the editor consults with an expert on AIC to help resolve this issue.

Because reviewer #1 remains uncertain about the AIC result, our response is that rather than continuing to discuss AIC, we can show other standard acceptable ways to compare models. One way is to simply compute the models on a subset of data (e.g., training data) and make predictions on held-out unseen data (e.g., test data), as is done with cross-validation techniques. Here we use 5-fold cross-validation to train the models on an 80% subset of the data, and then make predictions on the left-out 20% that it has not seen. The predictions that are made on this new held-out unseen data are then compared to the actual held-out unseen data and we can compute the prediction error for each model (e.g., mean squared prediction error, MSPE). In this context, it is simple to evaluate which model is best – the model that produces the lowest MSPE is the better model. We show that the subtype mode indeed produces lower MSPE than the case-control model for the DMN-OTC comparison (see second table below). Thus, we show with simple cross-validation that the subtype model for DMN-OTC makes better predictions on new unseen data than the case-control model.

Cross-validated (5-fold) mean squared prediction error (MSPE) for the subtype and case-control models.

DMN-OTCDMN-PVCDMN-DANSubtype Model0.04190.03280.0632Case-Control Model0.04300.03260.0634

We can also compute mean absolute percentage error (MAPE) as perhaps a more interpretable quantity than MSPE. MAPE is computed as the mean absolute value of percentage error and is computed as MAPE = mean(abs((A_i_ – P_i_)/A_i_)*100), where A_i_ is actual test data point i, P_i_ is predicted test data point i, and abs refers to the absolute value. The model with the lowest MAPE is the best model to choose. MAPE is computed for each cross validation fold, and then the average is reported across folds. The difference in MAPE values also allows for interpretability with regards to how much of a reduction in percentage error is gained by the better model.

Cross-validated (5-fold) mean absolute percentage error (MAPE) for the subtype and case-control models.

DMN-OTCDMN-PVCDMN-DANSubtype Model125.5452152.8099156.0619Case-Control Model135.2241152.1738152.8081

For DMN-OTC, MAPE was much lower for the subtype model (MAPE = 125.5452) compared to the case-control model (MAPE = 135.2241), indicating the subtype model reduced the absolute percentage of error by around 9.67%. However, for DMN-PVC and DMN-DAN, MAPE was lower for the case-control model (DMN-PVC subtype MAPE = 152.8099, case-control MAPE = 152.1738; DMN-DAN subtype MAPE = 156.0619, case-control MAPE = 152.8081), but in both cases the reduction in MAPE was <1%. In a revision we report cross validated MAPE to supplement the analyses showing ΔAIC.

3) I remain concerned about the weak theoretical framing surrounding the DMN-OTC results, a point that I highlighted in my initial review of this work. The DMN is typically considered a task-negative network, and understanding its link with task-positive OTC in neurotypical individuals or individuals with ASD is not mentioned by the authors. This is critical given the importance of the DMN-OTC results for this paper.

Reviewer #1 was concerned about the theoretical framing surrounding the DMN-OTC. Reviewer #1’s suggestion was that we introduce altogether new ideas that we did not originally have or justify at the beginning of this work, as part of the Introduction.

In our previous revision, we did our best to expand on our justifications and references in the Introduction, since the reviewer here is referring to previous comments about further expanding on ideas we had mentioned in the Introduction. However, it is clear from the reviewer’s comments here, that what they want is for us to address something they have in mind regarding task-negative versus task-positive networks. This was not part of our initial justifications. Going into detail like this that uses what is known about the results in an Introduction is an example of HARKing – Hypothesizing After Results are Known (Kerr, 1988). We therefore do not feel it is appropriate to cite this literature in the Introduction and, in the revised manuscript, we have not followed the reviewer’s suggestion here on this matter.

Reviewer #3:This reviewer thanks the authors for their efforts on this first revision of the manuscript. Several of my core concerns of the previous version of the manuscript revolved around details and inconsistencies in the statistical modeling.As one example from the revised manuscript:"Both GLM models were implemented with the lm function in R. In these GLMs the partial correlation for a particular component pair was the dependent variable.. […] The primary dependent variable of interest was the group variable."Here, the authors first mention the connectivity strengths as the dependent variable first, and then instead mention the group variable… This, and other remaining problems (explanation of bootstrapping etc.) make me doubt that the results and conclusion stand on solid ground – which I have already expressed in detail regarding the previous version of the manuscript.

This reviewer’s concern arose because we made a typo in our revision where we accidentally wrote “dependent” instead of “independent”.

Specifically, we wrote, “The primary *dependent* variable of interest was the group variable.” It should be, “The primary *independent* variable of interest was the group variable.”

This typo lead the reviewer to have concerns about the statistical modeling, but our modeling is solid and all the code for the analyses is all available on GitHub (https://github.com/mvlombardo/geoprefrsfmri) and was provided in the last revision. It is clear in the code how we did the analysis, including that “The primary independent variable of interest was the group variable.”

We are sorry for the typo and in a revised manuscript we have corrected this typo and have further made sure that all language referring to the analyses are clear.

Regarding the way bootstrapping is explained, our bootstrapping analysis helps us to generate a sampling distribution around the test statistic in question. The 95% CI bounds on this bootstrapped sampling distribution are reported, to give the reader an idea of how variable the test statistic could be with respect to its bootstrapped sampling distribution. This is consistent with what we have written in the revision.